

# A machine learning based approach for better prediction of fatigue life of offshore wind turbine foundations using smaller datasizes

Ahmed Mujtaba[1], Wout Weijtjens[1], Negin Sadeghi[1,2], and Christof Devriendt[1,2]

[1]OWI-Lab, AVRG, Vrije Universiteit Brussel (VUB), Brussels, 1050, Belgium
[2]24SEA, Brussels, 1000, Belgium

**Correspondence:** Ahmed Mujtaba (ahmed.mujtaba@vub.be)

**Abstract.**

As offshore wind turbine (OWT) foundations approach the end of their design life, the industry is increasingly focused on strategies for lifetime extension. As fatigue is the design driver for foundations of OWTs, reliable fatigue damage predictions are essential to support informed decisions for lifetime extensions. While simulation-based fatigue life reassessments are common, data-driven approaches using measured strain data have emerged as an alternative that can reduce modeling uncertainties. But, data-driven approaches face challenge as having access to strain data over the entire past lifetime is not an industry-standard. Often measurement campaigns are only kicked off when a lifetime extension is considered, thus limiting the availability of strain data. However, environmental and operational conditions (EOCs) of the wind turbines are usually recorded during the whole operational period. Using limited strain measurements and long term EOCs to estimate fatigue damage in unmonitored periods during the lifetime of the turbine requires temporal extrapolation techniques. Existing work on this topic presents several extrapolation methods, including linear time-based extrapolation, binning based on correlations between EOCs and average damage, and machine learning (ML) models. The accuracy of these methods depends on factors such as the selected EOC parameters, the duration and starting point of available strain data, the power rating and type of the wind turbine, as well as the type and architecture of the extrapolation model used. This study presents a novel machine learning based extrapolation model using random forest (RF) for temporal extrapolation of strain measurements. A comparative analysis of novel RF model with previously identified binning models is presented. The extrapolation performance is validated using five years of measured strain, SCADA, and wave data from a 3 MW and a 9 MW OWT installed on monopile foundations in the Belgian North Sea. Using a sliding window approach on the available monitoring data, we estimate and compare the statistical uncertainty in fatigue life predictions of various extrapolation models. The results indicate that wave parameters play a more significant role in fatigue prediction for larger turbine of 9 MW compared to smaller one of 3 MW power rating. For limited data sizes, less than 12 months, the proposed RF model demonstrates superior performance, offering more reliable fatigue life predictions with reduced statistical uncertainty. However, for longer datasets, greater than 12 months, the performance advantage of RF model over binning methods becomes less pronounced.



## 1 Introduction

As offshore wind farms mature, extending the operational lifetime of wind turbines has become a pressing concern for the wind energy industry. According to WindEurope (2017), a substantial share of the installed wind capacity in the European Union is expected to reach the end of its design life between 2020 and 2030. Decommissioning these assets without extending their service would hinder progress toward the EU's 2030 target of achieving 50% of electricity from renewable sources.

Therefore, lifetime extension, alongside repowering and new installations, is vital for meeting long-term sustainability goals. As highlighted by Shafiee (2024), extending the operational life of OWTs offers significant economic and environmental benefits, including reduced Levelized Cost of Energy (LCOE) and lower emissions.

Fatigue is a governing factor in the structural design of wind turbine support structures, which are primarily optimized for dynamic—rather than static—loading conditions (Sparrevik (2019)). Monopile foundations are typically designed for a

service life of 20–25 years. However, findings from some structural health monitoring (SHM) campaigns suggest that actual fatigue loads may be lower than anticipated, revealing unexploited fatigue capacity and motivating detailed reassessments for lifetime extension purposes (Tewolde et al. (2018)). For example, in the case of lifetime extension of the Samsø offshore wind farm in Denmark, the Danish Energy Agency required detailed fatigue reassessments using updated load conditions to support lifetime extension permits (Buljan (2025)). International guidelines for lifetime extension, such as those outlined in DNV-ST-

0262 (2016), recommend fatigue life reassessments based on updated models that incorporate measured environmental and operational data. While traditional fatigue reassessments rely heavily on simulation-based models, the availability of measured strain, as typically provided by SHM systems, and long term EOC data opens the door for data-driven alternatives that can reduce uncertainties associated with initial design assumptions in the simulation based models (Kinne and Thöns (2023)).

To support lifetime assessments, two primary approaches exist: simulation-based reassessments using updated models, and

data-driven methods using measured strains. The latter mitigates modeling assumptions but introduces new challenges— such as high cost of installation and maintenance of strain gauges on OWTs (Bezziccheri et al. (2017)) and limited data availability (Pacheco et al. (2023)). Strain gauges are typically installed at a few critical locations, requiring spatial extrapolation to predict strains at locations where no direct measurements are available. Strain measurements are limited in duration as having strain data over the entire past lifetime is not an industry-standard and because of the challenges associated with traditional sensor

deployment in harsh marine environments, such as sensor failures due to corrosion caused by saltwater and humidity, and high costs related to complex installation logistics (Ángel Encalada-Dávila et al. (2025)). Often measurement campaigns are only started when a lifetime extension is considered, thus limiting the availability of strain data. On the other hand, a Supervisory Control and Data Acquisition (SCADA) system is typically installed on OWTs to capture and record EOCs such as wind speed, power, rotational speed etc., to enable the wind farm operators to track and control the turbines performance in real

time (Moynihan et al. (2024)). The available data paves the way for temporal extrapolation techniques that use limited strain measurements and long term EOCs to estimate fatigue damage in unmonitored periods during the lifetime of the turbine.

Prior research has extensively explored extrapolation of strain and load data for offshore wind turbines, both spatially, to uninstrumented locations, and temporally, beyond the measurement window. Spatial extrapolation studies include those





targeting different positions on the same turbine (Ziegler et al. (2019); Moynihan et al. (2024); Fallais et al. (2025); Ziegler et al. (2017); Zou et al. (2023); Zhang et al. (2024); Simpson et al. (2025); Ángel Encalada-Dávila et al. (2025)), as well as farm-wide extrapolation approaches where measurements from one or a few instrumented turbines, called fleet leaders, are extended to others across the farm (de N Santos et al. (2024); Weijtens et al. (2016); Noppe et al. (2020); Pacheco et al. (2023)).

Temporal extrapolation methods vary in complexity, ranging from simple linear techniques to binning strategies that relate fatigue damage to EOCs, and more advanced ML approaches. Notable studies have assessed the accuracy of these methods using various datasets (Hübler and Rolfes (2022); Ziegler and Muskulus (2016); Weijtens et al. (2016)). For example, Hübler et al. (2018) evaluates linear extrapolation, also called 0-dimensional (0d) binning, 1d binning, where 10-minute fatigue damage is split into wind speed bins, and 2d binning, splitting fatigue damage into wind speed and wind direction bins using strain measurements from a 3MW OWT installed on a monopile. Hübler et al. (2018) concludes that strain measurements of 9 to 10 months provide a representative and unbiased dataset with 1d binning using wind speeds giving the most reliable fatigue life estimations. Hübler and Rolfes (2022) studies the performance of multi-dimensional binning extrapolation, Artificial Neural Networks (ANN) and Gaussian Process Regression (GPR) trained using multiple 1-year periods of measured strains. In Hübler and Rolfes (2022), binning approaches using wind speed correlations provide best results. Pacheco et al. (2022) summarizes the steps for calculating and extrapolating fatigue damage in wind turbines using strain measurements. Apart from estimating fatigue life of instrumented turbine, Pacheco et al. (2022) concludes that the results form instrumented turbines can be extrapolated to uninstrumented turbines in the same wind farm. Pacheco et al. (2023) uses a damage capture matrix formed using 2d binning in wind speed and turbulence intensity, to extrapolate strain measurements from one turbine to other turbines in an onshore wind farm. Pacheco et al. (2023) recommends using 1-year monitoring period and concludes that the uncertainties in extrapolations decrease with increasing monitoring period.

The rapid development and advances in computer science and its applications in wind energy pose machine learning models as a favorable option for temporal extrapolation of fatigue damage in wind turbines (He et al. (2022); Raju et al. (2025)). Literature focuses on using ML models as surrogates trained on simulated data to predict fatigue loads on each turbine in a wind farm (Bossanyi (2022); Gasparis et al. (2020); Singh et al. (2022)). There is limited literature on use of ML models trained on strain measurements and used for fatigue damage predictions such as de N Santos et al. (2023) uses Physics-guided learning of neural networks trained on 9 months of strain measurements for long-term fatigue damage estimation using SCADA and acceleration data. In wind energy applications, Random Forest model is proved to be a powerful ensemble learning method, widely used for regression and classification tasks. RF models adapt excellently to high-dimensional data and large-scale datasets, and are robust in dealing with missing values and unbalanced datasets (Karadeniz (2025)). For example, Karadeniz (2025) compares RF, Long Short-Term Memory Network (LSTM) and and Gated Recurrent Unit (GRU) for predicting Total Harmonic Distortion Voltage (THDV) of offshore wind farms to conclude that RF models outperforms LSTM and GRU in predicting THDV with lowest root mean squared error (RMSE). Similarly, Zhou et al. (2016) uses RF regression model to predict short term power production of a wind farm and Rouholahnejad and Gottschall (2025a) uses RF model to extrapolate near surface wind speed up to 200m.





Literature review highlights that the sensitivity of ML and binning based extrapolation models to different turbine power ratings remains poorly understood. Many studies assume homogeneity in turbine design and operating conditions, often focusing
on a single turbine type and neglecting spatial variability across the farm (Bouty et al. (2017)). Consequently, it is unclear how extrapolation performance varies across turbines with different power ratings, or environmental exposures.

Moreover, validation of these models is often limited to short timescales (e.g., several months), which restricts confidence in their long-term predictive capability. While several binning-based and regression-based methods have been proposed (Ziegler et al. (2017); Pacheco et al. (2023); Sadeghi et al. (2023b)), their comparative effectiveness and robustness under varying data
availability, input features, and turbine power ratings are not yet well established. Especially in discussions of lifetime extension for aging turbines, methodologies that require shorter measurement campaigns are advantageous. This raises key questions like: What is the minimum duration of the monitoring period required for statistically reliable fatigue life predictions? and Can machine learning models reduce this monitoring period without sacrificing accuracy or increasing uncertainty?

Despite the increasing attention to data-driven extrapolation, key gaps remain in the literature:

– A systematic comparison of binning and machine learning, specifically random forest, models for fatigue life prediction across multiple turbine sizes is missing.

    – The relative importance of SCADA and wave parameters selection for these models to predict fatigue damage has not been thoroughly assessed for turbines of different power ratings.

    – The ability of these models to predict fatigue damage across different directions (fore-aft, side-side, or single-sensor
measurements) has not been thoroughly evaluated for turbines of varying capacities.

    – The sensitivity of these models to data availability, including variations in dataset size and measurement start time, is underexplored.

This study addresses these gaps by first introducing a novel ML based extrapolation model using random forest for temporal extrapolation of strain measurements. This model is compared with state-of-the-art temporal extrapolation techniques for
fatigue life prediction and validated using five years of measured strain, SCADA, and wave data from two offshore wind turbines: a 3 MW and a 9 MW turbine, both installed on monopiles in the Belgian North Sea. We analyze the influence of dataset size, measurement start time, model dimensionality, and feature selection on extrapolation accuracy. Multiple configurations of random forest models are tested, including recursive feature elimination with cross-validation (RFECV) and state-specific modeling, to explore trade-offs between model complexity and predictive performance.
The findings provide new insights into the reliability, convergence behavior, and practical limitations of data-driven fatigue extrapolation models, offering valuable guidance for their deployment in support of lifetime extension assessments.

The remainder of the paper is structured as follows: It begins with the Objective section , leading into the Measurement Setup for strain, collecting SCADA, and wave data. The paper then explains the Methodology section. This section elaborates on model development with a focus on feature selection, binning, and the RF model, outlines the technique for statistical
uncertainty estimation applicable to these extrapolation models and details the approach for damage extrapolation used in




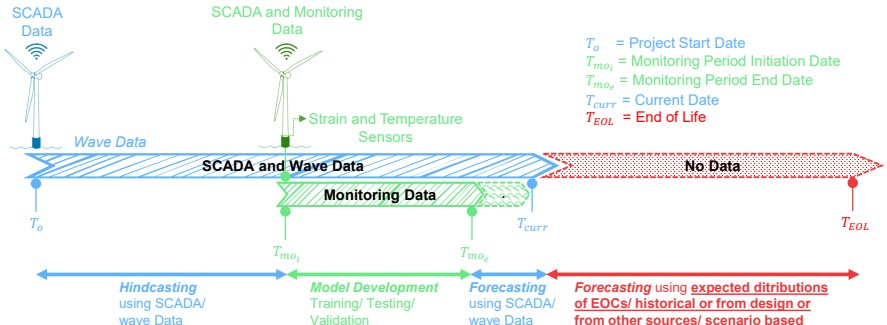

**Figure 1.** A timeline showing model developed using monitoring data used for hind-casting and forecasting using SCADA-wave data where available, and forecasting in future using expected distributions of EOCs

estimating fatigue life. The paper then concludes with a presentation of Results of fatigue lifetime predictions in various directions along with discussions of the results.

## 2 Objective

Consider initiating a strain measurement campaign aimed at reassessing the fatigue life of OWT with the goal of extending their lifetime. How long should measurements be conducted to ensure a reliable estimate of fatigue life? The growing number of aging OWTs demands fast, reliable, and data-efficient methods to accurately predict their end-of-life (EOL). Addressing this question is important since it can result in reduced costs for prolonged measurement campaigns and more time-efficient fatigue life estimates. Precise predictions of fatigue life are essential for making informed lifetime extension decisions; however, this requires balancing the duration of measurement campaigns with both the financial costs of data acquisition and the accuracy of the predictions.

Figure 1 illustrates the typical scenario encountered in fatigue life estimation. A strain sensor is installed at a later stage of turbine's operation, for example when lifetime extension is considered, initiating a monitoring period between $T_{mo_i}$ and $T_{mo_e}$. Meanwhile, SCADA and wave data are usually available from the turbine's commissioning at $T_o$. During the monitoring period $\Delta T_m = T_{mo_e} - T_{mo_i}$, extrapolation models are trained using measured strain and corresponding SCADA/ wave data. These models are then used to estimate cumulative fatigue damage up to the projected end-of-life time $T_{EOL}$, leveraging either historical SCADA/wave records or design-based EOC distributions. Such models can also facilitate scenario-based forecasting of fatigue damage under varying operational profiles.

This study presents a ML based extrapolation technique using RF models and systematically investigates the uncertainty in fatigue life predictions under varying monitoring periods and starting points. Given the seasonality of EOCs, the timing and length of the data window can significantly influence model robustness. In addition, factors such as binning strategies (e.g.,



dimensionality, bin resolution, empty-bin handling) and feature selection techniques in ML-based models play a critical role in the extrapolation accuracy.

Using long-term strain, SCADA, and wave measurements from two OWTs installed on monopiles—this study aims to:

- Quantify the minimum monitoring duration $\Delta T_m$ required for reliable fatigue life predictions using ML and binning extrapolation models.


- Develop robust estimates of turbine end-of-life $T_{\mathrm{EOL}}$ using extrapolation models based on measured strain and EOCs data.

- Estimate uncertainty bounds associated with $T_{\mathrm{EOL}}$ under different modeling approaches.

- Evaluate the potential of machine learning models, Random Forests, to reduce required monitoring durations and improve prediction reliability, in comparison to traditional extrapolation methods such as linear and binning-based models.


- Compare the performance of different extrapolation models across varying offshore wind turbine power ratings, 3 MW and 9 MW, to assess model reliability.

- Compare the ability of these models to predict fatigue damage across different directions, fore-aft, side-side, or single-sensor measurements, and for turbines of different power ratings, 3 MW and 9 MW.

## 3 Measurement Setup

Table 1 provides an overview of the parameters used in this study along with the data availability. 5 years of strain measurements, SCADA and wave data from a 3MW and 9MW OWT are used. The strain measurement setup consists of six circumferential strain gauges installed on the transition piece (TP) near the tower-TP interface. Figure 2 shows the sensor locations along the circumference and the dominant wind direction of 230° for both turbines. The closest sensor to the dominant 165 wind direction is at 205° for 3MW OWT and 230° for 9MW OWT.

### 3.1 Strain data

The data from six circumferential strain sensors is pre-processed using the steps shown in Fig. 3. The strain measurements are converted to stress using Hooke's Law. The stress measurements per sensor are then used to calculate stresses in fore-aft (FA) and side-side (SS) directions which are de-facto standard loading directions in wind turbines. Each 10 minute signal is 170 processed using rainflow cycle counting to get stress range histogram.

Stress range histograms or cycle count matrices are scaled using a combination of static extrapolation, safety, and correction factors. The static extrapolation factor is employed to project the measured stress signals onto fatigue-critical structural locations, without modeling structural dynamics. This factor is computed as the product of two ratios: (i) the bending moment ratio between the extrapolated and measured locations, obtained from bending moment diagrams either in design documentation or 175 reanalysis; and (ii) the section modulus ratio at the two locations.





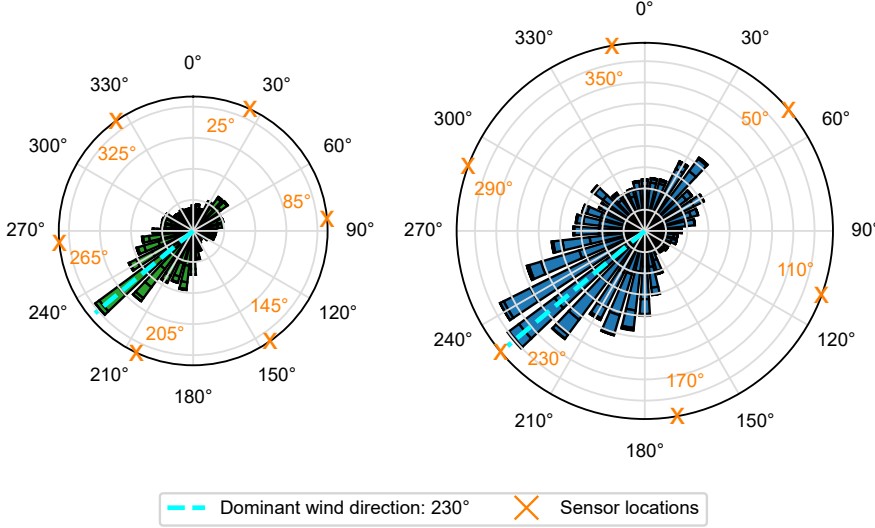

**Figure 2.** Sensors location and windrose showing dominant wind direction for a 3MW (left) and 9MW (right) OWT

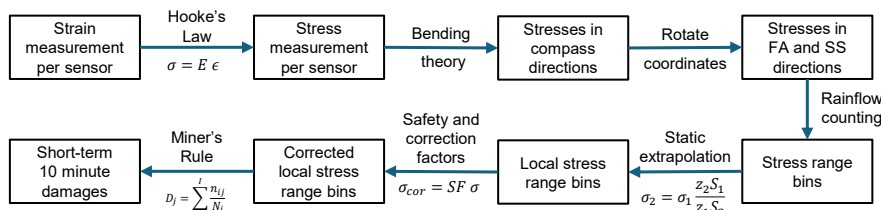

**Figure 3.** Strain data preprocessing to calculate short-term 10 minute damages in FA and SS directions (Image reproduced from Hübler et al. (2018))

Additional correction factors include a stress concentration factor (SCF), a material safety factor (MSF) usually defined in guidelines such as IEC-61400-1 (2019), and a thickness correction factor following DNV-RP-C203 (2024) guidelines to account for the size effects. These factors are applied to generate a corrected stress range histogram for fatigue damage analysis. However, the exact numerical values of these scaling factors are not essential for the temporal extrapolation process and do not

affect the conclusions of this paper ( Hübler and Rolfes (2022)).

Subsequently, the fatigue damage corresponding to each stress range bin is computed using the appropriate S-N (Wöhler) curve described by Basquin Equation 1 (BASQUIN (1910)), where $log(\bar{a})$ is the intercept term and m is the slope (or Wöhler exponent) of the SN curve, $\Delta\sigma$ is the stress range and $N$ is the number of cycles to failure.

$$\log(N) = \log(\bar{a}) - m \cdot \log(\Delta\sigma) \tag{1}$$



The individual damage contributions are calculated and accumulated over time using Palmgren-Miner's linear damage rule (Miner (1945)) to estimate the fatigue damage per 10-minute interval, as shown in Equation 2.

$$D = \sum_{i=1}^{k} \frac{n_i}{N_i} \tag{2}$$

where:

- $n_i$ is the number of cycles in the $i^{\text{th}}$ stress range bin,

- $N_i$ is the number of cycles to failure at the $i^{\text{th}}$ stress range (obtained from the S–N curve),

- $k$ is the total number of stress range bins.

10-minute fatigue damage is used as the extrapolation models target variable. Strain data processing, cycle counting, stress range corrections, fatigue damage calculation and accumulation are done using open source python package py-fatigue (D'Antuono et al. (2023)). In this study, fatigue life is estimated using three approaches based on damage values:

- **Single Sensor Damage:** Fatigue life is estimated using damage calculated from the sensor nearest to the dominant wind direction. This approach bypasses the rotation of stress measurements and directional compass transformation (see Fig. 3), applying rainflow counting directly to the raw stress time series.

- **Fore-Aft (FA) Damage:** It is assumed that the turbine faces fore-aft damage conditions throughout its lifetime. This method is conservative and likely underestimates fatigue life, since FA damage state is not constantly maintained over time.

- **Side-Side (SS) Damage:** It is assumed that only side-side damage prevail throughout turbine lifetime. This approach allows estimation of life consumption if SS damage states were persistent.

The methodology used to extrapolate damage predictions and estimate fatigue life ($T_{\text{EOL}}$) is described in detail in Section 4.3.

### 3.2 SCADA and wave data

The SCADA system provides essential turbine parameters including wind speed, yaw angle, pitch angle, rotor power output, and rotational speed. The corresponding symbols used for these variables throughout this study are summarized in Table 1. 10-minute mean statistics of these parameters are utilized as inputs for extrapolation models development.

Turbine operational states defined based on the SCADA including nominal, idling, highwind, abnormal (only for 9MW OWT), curtailed and invalid are incorporated into both binning and ML based extrapolation models. *Invalid* operational states refer to intervals lacking valid SCADA-derived statistics, typically involving transient events such as rotor start-up or shut-down. For the 9 MW OWT, an additional *abnormal* operational state is identified, associated with curtailed power output. The





**Table 1.** Overview of SCADA, wave and strain parameters for 3MW and 9MW OWT, their sources, duration and availability factors

<table>
<tr><td></td><td>**Sensor**</td><td>**Statistic**</td><td>**Variable**</td><td>**Symbol**</td><td>**Units**</td><td>**Data Availability (%) [3MW]***</td><td>**Data Availability (%) [9MW]****</td></tr>
<tr><td rowspan="12">**Inputs**</td><td rowspan="7">**SCADA**</td><td rowspan="7">10 minutes mean</td><td>Windspeed</td><td>WS</td><td>m/sec</td><td>99.87</td><td>98.22</td></tr>
<tr><td>Yaw Angle</td><td>WD</td><td>Deg</td><td>99.87</td><td>98.48</td></tr>
<tr><td>Pitch Angle</td><td>PI</td><td>Deg</td><td>99.87</td><td>97.81</td></tr>
<tr><td>Power</td><td>PW</td><td>kW</td><td>99.74</td><td>98.3</td></tr>
<tr><td>Rotational Speed</td><td>RPM</td><td>rpm</td><td>99.87</td><td>98.62</td></tr>
<tr><td>Turbulence Intensity</td><td>TI</td><td>-</td><td>98.66</td><td>98.22</td></tr>
<tr><td>Operational State†</td><td>OS</td><td>-</td><td></td><td></td></tr>
<tr><td rowspan="4">**Wave Data**</td><td rowspan="4">30 minutes mean</td><td>Mean Wave Height</td><td>WH</td><td>cm</td><td>99.98</td><td>99.98</td></tr>
<tr><td>Mean Wave Period</td><td>WP</td><td>sec</td><td>99.98</td><td>99.98</td></tr>
<tr><td>High Frequent Wave Direction</td><td>HF</td><td>Deg</td><td>99.96</td><td>99.97</td></tr>
<tr><td>Low Frequent Wave Direction</td><td>LF</td><td>Deg</td><td>99.96</td><td>99.97</td></tr>
<tr><td></td><td>5 minutes mean</td><td>Tidal level***</td><td>TL</td><td>cm</td><td>100</td><td>100</td></tr>
<tr><td rowspan="3">**Fatigue Target**</td><td rowspan="3">**Strain Gauges**</td><td rowspan="3">10 minutes fatigue damage</td><td>Single Sensor Damage</td><td>-</td><td></td><td>96.92</td><td>97.78</td></tr>
<tr><td>Fore-Aft Damage</td><td>FA</td><td>-</td><td>96.77</td><td>96.02</td></tr>
<tr><td>Side-Side Damage</td><td>SS</td><td>-</td><td>96.77</td><td>96.02</td></tr>
</table>

\* 5 years of monitoring and SCADA data used 2020-01-01 till 2024-12-31

\*\* 4.67 year of monitoring and SCADA data used 2020-05-01 till 2024-12-31

\*\*\* Tidal levels are taken from Ostend harbour - Tide due to lower data availability for Wandelaar Measuring pile [all other wave data taken from Thorntonbank South Buoy]

†Operational States for 3MW 'nominal', 'invalid', 'idling', 'highwind'

Operational States for 9MW 'nominal', 'invalid', 'idling', 'highwind', 'abnormal'

*nominal* power generation and *idling* (Parked) operational states are defined consistently across both the 3 MW and 9 MW OWTs.

Turbulence intensity (TI) is computed for each 10-minute interval using the ratio of the standard deviation to the mean wind
speed, defined as:



$$TI = 100 \times \left( \frac{\sigma_{\text{wind speed}}}{\mu_{\text{wind speed}}} \right) \tag{3}$$

Wave and tidal data are typically not included in wind farm SCADA datasets. For this study, publicly available environmental data from the Meetnet Vlaamse Banken (MVB (2025)) covering the Belgian part of the North Sea are used. Meetnet Vlaamse Banken comprises wave buoys and measurement piles deployed across the North Sea, providing measurements of wind, wave,

and tidal conditions. Reported wave characteristics include wave height, 10% highest waves, height of waves with period greater than 10 seconds, high frequent wave direction, low frequent wave direction, average wave period and tidal levels. These measurements are generally reported with sampling frequency of 15–30 minutes and are resampled with interpolation to align with the 10-minute resolution of SCADA measurements from OWTs.

Wave data from the Thorntonbank South buoy are selected for analysis, as this station offers high data availability during the

study period and is geographically closest to the considered wind farms having a distance of 18.5 km from 3 MW OWT and 7.8 km from 9 MW OWT. Although this location does not provide tidal data, tidal measurements from the Wandelaar station were evaluated but excluded due to a low data availability during the time of interest. Instead, tidal data from the Oostende station are used, as they align well with wave radar observations at one of the wind farms analyzed of which a small window was available for validation. A potential time offset exists between wave events at the OWT site and the measurement station

due to spatial separation. A simplified time-shift estimation, assuming deep-water wave propagation and direct travel from the buoy to the turbine location, yields a maximum offset of approximately 40 minutes. However, no explicit correction for wave time offset is applied in this study.

## 4 Methodology

Figure 4 illustrates an overview of the methodology, where various extrapolation models are applied to different monitoring

period sizes and starting points. A selected monitoring window $\Delta T_m$, with varying durations and starting points, is used to train extrapolation models. These models predict 10-minute fatigue damage based on SCADA/ wave parameters which is used to estimate fatigue life $T_{\text{EOL}}$ over the full operational timeline.

### 4.1 Model Development

Two modeling approaches are investigated for fatigue life extrapolation: (1) multi-dimensional binning (Pacheco et al. (2022);

Hübler et al. (2018); Hübler and Rolfes (2022); Sadeghi et al. (2023b)), and (2) Random Forest Model. The Random Forest model has been selected owing to its robustness and better performance relative to other machine learning models (Karadeniz (2025); Rouholahnejad and Gottschall (2025a)). This paper focuses on the fundamental question as to whether the deployment of ML models offers any significant benefits instead of comparing the performance of various ML models.

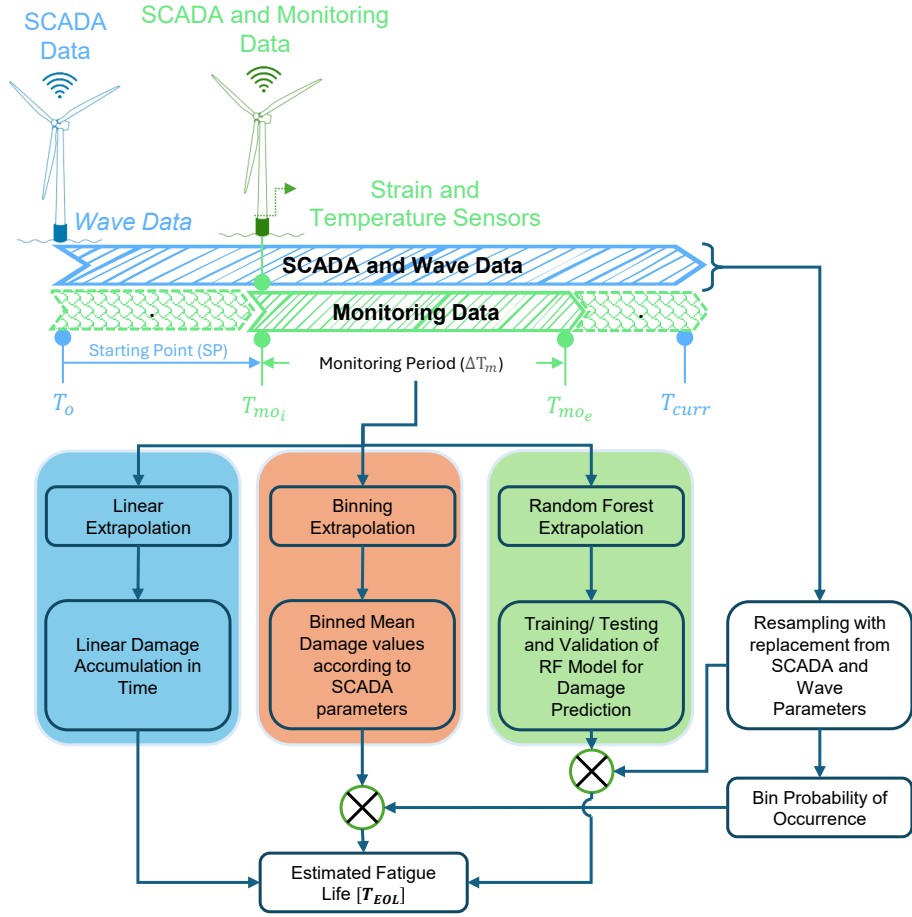

**Figure 4.** 5 years of data $T_{curr} - T_o$ is split into a sliding window of variable size $\Delta T_m$ and starting point $SP$ used for models development. Fatigue life $T_{EOL}$ is estimated using complete SCADA and Wave data.

10 minute SCADA and wave statistics are used as input parameters and 10-minute fatigue damage is the target variable as summarized in Table 1. Extrapolation models are evaluated under varying monitoring durations and feature subsets to assess their robustness and prediction accuracy.

#### 4.1.1 Feature Selection

Feature selection is a crucial step, especially in fatigue modeling where input parameters, to be precise the SCADA and wave data, may vary in importance depending on the operational state of the turbine and the size of the training dataset. Feature selection is performed for both the binning and RF models to enable a fair comparison by using the same set of input features. Four approaches to feature selection are employed:





1. **Random Forest Feature Importance:** A filter-based method is applied using the built-in feature importance metric from a Random Forest model trained on 12 months of monitoring data. The top five features, capturing over 80% of cumulative importance, are selected. This 5-feature limit enables a fair comparison between a 5-dimensional binning approach and RF model using five features, denoted by RF-5D. Feature selection is performed separately for different fatigue targets, fore-aft, side-side, and single sensor damage, since the relevance of features differs for these damage types. Categorical variables, such as turbine operational states, are transformed using one-hot encoding to enable their use in tree-based models.

2. **Global RFECV:** A recursive feature elimination with cross-validation is applied globally, i.e., across all operational states, using the same 12-month dataset. This method, referred to as Global RFECV, aims to identify a reduced yet effective feature subset optimized for overall model performance.

3. **State-Specific RFECV:** To account for opertaional state-dependent fatigue behavior, RFECV is applied individually to each turbine operational state, generating unique feature subsets per state. Separate Random Forest models are trained for each state, significantly increasing the number of models but allowing for tailored fatigue prediction.

4. **Full-Feature Baseline:** A baseline model using all available features is also developed to serve as an upper-bound benchmark, against which performance of RF-5D and RFECV-based models is compared.

#### 4.1.2 Binning Extrapolation

In the binning approach, fatigue damage values are discretized based on selected EOC parameters. A single statistic, typically mean, is computed to represent all damage in each bin, resulting in a multi-dimensional damage matrix. To estimate long-term fatigue life, EOC parameters are sampled from extended operational periods to determine bin probabilities. These probabilities, representing the occurrence likelihood of EOCs, are combined with the damage matrix to compute the expected fatigue damage. To account for varying turbine operational states, the monitoring period $\Delta T_m$ is first segmented by operational states, and a separate damage matrix is constructed for each state. This allows the binning approach to accurately reflect state-dependent fatigue behavior.

In this paper, the dimensionality of binning is increased from 1D to 5D. In 1D binning, only wind speed is used if wind speed is in the selected features. This is because Hübler and Rolfes (2022) suggests that binning approaches using wind speed correlations provide best results. For 2D binning, wind speed along with the parameter with the highest importance, as determined by feature_importances_ scores from a random forest model, is used for discretization. For higher dimensions, the top-ranked features are sequentially added—e.g., 3D binning uses wind speed and the top two features, 4D binning uses wind speed and the top three features, and so on, up to 5D.

Explicit bin size optimization is not performed in this study. A critical challenge in multi-dimensional binning is the treatment of empty bins, which can arise due to data sparsity in high-dimensional feature spaces. For example in 3D binning based on wind speed, wind direction, and power, certain combinations of bin values may not be observed within the monitoring



period, particularly when the monitoring duration is short, resulting in empty bins. Several strategies have been proposed in the literature to address this issue.

For instance, Noppe et al. (2020) suggests filling empty bins with the maximum damage value observed in adjacent or neighboring bins, thereby maintaining conservative fatigue estimates. Pacheco et al. (2023) proposes a more sophisticated interpolation approach, fitting a three-dimensional surface to the damage matrix to estimate missing values. Simpler and more commonly applied methods include filling empty bins with the mean damage from samples in the same wind speed bin, the 90th percentile, or the maximum damage. For example, Hübler et al. (2018) suggests filling the empty bins with highest statistical damage value of the same wind speed bin but concludes that in this strategy bins are filled up conservatively leading to reduced lifetimes. However, Hübler and Rolfes (2022) adopts filling empty bins with the maximum value observed in the surrounding bins as suggested by Noppe et al. (2020).

In this study, the performance of bins filling with mean, 90th percentile and maximum damage value of the same wind speed bin is evaluated (see Fig. A1, Appendix A). Extreme values—such as the 90th percentile or maximum—tends to significantly overestimate fatigue damage, thus underestimating fatigue life. In contrast, mean-value bin filling yields more consistent and realistic fatigue predictions and is therefore adopted throughout this work.

### 4.1.3 Random Forest Model

Random Forest is an ensemble learning method widely applied to regression and classification tasks. It enhances prediction accuracy and stability by constructing multiple decision trees and aggregating their outputs. Each tree is trained on a bootstrap sample, created by sampling with replacement from the original dataset, and, at each split, a random subset of features is considered to increase model diversity and reduce overfitting risk. RF models demonstrate strong adaptability to high-dimensional and large-scale datasets, as well as robustness in handling missing values, unbalanced data, noise, and outliers. Furthermore, RF inherently provides feature importance rankings, offering valuable insight into the relative contribution of input variables to the prediction outcome (Karadeniz (2025)).

The RF regression model is used to predict 10-minutes fatigue damage based on input SCADA and wave parameters. To handle categorical variables, turbine operational states, as described in Section 3.2, are transformed using one-hot encoding, converting each category into a separate binary feature. The target variable, 10-minutes damage, is normalized using a Min-MaxScaler to scale values between 0 and 1, facilitating stable model training.

RF model is trained using open source python package Scikit-learn (Pedregosa et al. (2011)) and a bayesian optimization approach is employed for tuning the hyperparameters of the RF model using Hyperopt package in python (Bergstra et al. (2013)). The hyperparameter search space includes the number of estimators (`n_estimators` (10, 300)), maximum depth of trees (`max_depth` (2, 15)), minimum samples required to split a node (`min_samples_split` (2, 15)), and the minimum number of samples required at a leaf node (`min_samples_leaf` (1, 7)).

Bayesian optimization is performed with a maximum of 50 evaluations (`max_evals=50`) to identify the optimal set of hyperparameters. Each monitoring period $\Delta T_m$ is split into 80% training and 20% testing subsets to evaluate the model performance and generalizability.





**Table 2.** Model development for statistical uncertainty estimation using different data sizes and starting points

| $\Delta T_m$ [**Months**] $\rightarrow$ <br> **Starting Point [SP]** $\downarrow$ | 1 | 2 | 3 | 6 | 9 | 12 | 15 | 18 | 21 | 24 |
|---|---|---|---|---|---|---|---|---|---|---|
| 1 | $M_{1-1}$ | $M_{2-1}$ | | | | | | | | |
| 2 | $M_{1-2}$ | $M_{2-2}$ | | | | | | | | |
| ... | | | | | | | | | | |
| 20 | $M_{1-20}$ | $M_{2-20}$ | | | | | | | | |

## 4.2 Statistical uncertainity estimation

Statistical uncertainty in this paper refers to the scatter in fatigue life predictions by models trained using $\Delta T_m$, but the starting
point of $\Delta T_m$ could be anywhere within $T_{curr} - T_o$ as shown in Fig. 4. This aims to answer the question, how much does the
starting point $SP$ of the measurement campaign influence the outcome on $T_{EOL}$ after a campaign of $\Delta T_m$ months.

To achieve this, the monitoring period $\Delta T_m$ is varied from 1 month to 24 months to study the effect of varying data size
on the model predictions. For each size of monitoring period $\Delta T_m$, the starting point (SP) (Figure 4) is randomly varied to 20
different positions to study the effect of starting point of monitoring data. This gives 20 different models for each monitoring
period $\Delta T_m$ as shown in Table 2. An extrapolation model trained with monitoring period $\Delta T_m$ and with a random starting
point SP is denoted as $M_{\Delta T_m - SP}$. Each model predicts 10-minute fatigue damage which is accumulated to a fatigue life thus
giving 20 different predictions for each monitoring period $\Delta T_m$.

## 4.3 Damage extrapolation for $T_{EOL}$

Damage extrapolation to estimate $T_{EOL}$ is done by resampling long-term SCADA and wave parameters from $T_{curr} - T_o$,
including data from both within and before and after the monitoring window, with replacement to create a synthetic dataset
for 20 years, reflecting the full lifetime, which are used to predict fatigue life. Each trained model is used to predict 10-minute
damages using this synthetic dataset. The predicted damages are accumulated to get yearly accumulated fatigue damage which
is used to calculate mean yearly damage as shown in Fig. 5. Mean yearly damage is calculated using equation 4.

$$\text{Mean Yearly Damage} = \frac{1}{20} \left( \sum_{j=1}^{20} \sum_{i=1}^{52560} d_{ji_{(10-\text{minutes})}} \right) \tag{4}$$

Where $i$ is the number of 10 minute samples in each year and $j$ is the total number of years. $d_{ji}$ is the $i^{th}$ 10-minute fatigue
damage for $j^{th}$ year.

The mean yearly damage is used to calculate fatigue life $T_{EOL}$ as shown in Fig. 5. The fatigue life $T_{EOL}$ is calculated using
equation 5.





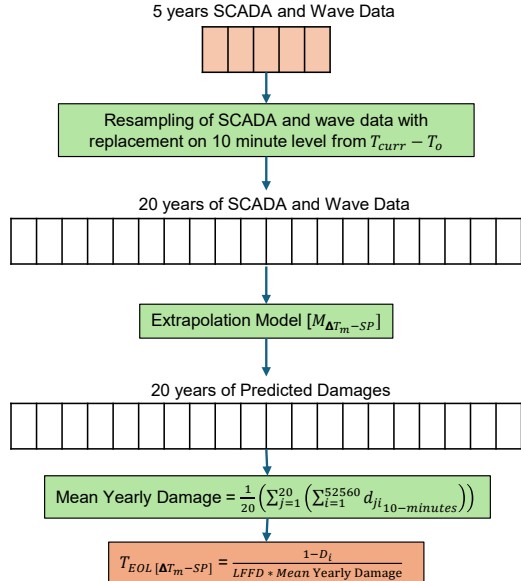

**Figure 5.** Extrapolation of 10 minute damages to calculate end of life of OWT

$$T_{\text{EOL } [\Delta T_m - \text{SP}]} = \frac{1 - D_i}{LFFD_{factor} \times \text{Mean Yearly Damage}} \tag{5}$$

$D_i$ is the initial damage including but not limited to pile driving damage, transportation damage, fatigue damage of the monopile foundations before start of normal operation, before turbine mounting and powering. In this study $D_i$ is considered as zero as it does not affect the models comparison. In line with findings in Sadeghi et al. (2023a) a low frequency fatigue factor $LFFD_{factor}$ can be included to account for long term cycles, that don't close within the default 10 minute window, however for simplicity a LFFD=1 is used.

## 5    Results

This section presents a comparative evaluation of the binning approach with varying dimensionality and Random Forest models across different target variables—namely, single-sensor, FA, and SS damage. The predicted fatigue life for single sensor and FA is normalized between (0.1–1) using Equation 6. The scaling of SS fatigue lifetimes is done with the same scale as of FA fatigue life to retain their relative magnitudes for comparison.

$$\bar{T}_{\text{EOL}} = \left( \frac{T_{\text{EOL}} - T_{\text{EOL,min}}}{T_{\text{EOL,max}} - T_{\text{EOL,min}}} \right) \cdot (\text{Scale}_{\max} - \text{Scale}_{\min}) + \text{Scale}_{\min} \tag{6}$$

where:





**Table 3.** Overview of selected features for different target variable for 3MW and 9MW OWT

|  | 3MW OWT | | | 9MW OWT | | |
| --- | --- | --- | --- | --- | --- | --- |
| **Variable** | Single sensor 205° | Fore-Aft | Side-Side | Single sensor 230° | Fore-Aft | Side-Side |
| Windspeed | 5 | 3 | 1 | 5 | 3 |  |
| Yaw Angle | 1 |  |  | 2 |  | 2 |
| Pitch Angle | 3 | 2 | 4 | 1 | 2 |  |
| Power | 2 | 5 | 5 |  |  |  |
| Rotational Speed |  | 1 |  |  | 4 |  |
| Turbulence Intensity | 4 | 4 | 2 | 4 | 1 |  |
| Operational State* | ✔ | ✔ | ✔ | ✔ | ✔ | ✔ |
| Mean Wave Height |  |  | 3 | 3 | 5 | 1 |
| Mean Wave Period |  |  |  |  |  | 4 |
| High Frequent Wave Direction |  |  |  |  |  | 5 |
| Low Frequent Wave Direction |  |  |  |  |  |  |
| Tidal level |  |  |  |  |  | 3 |

\* Operational state is deliberately included in all models (a 5D model contains 5-EOC parameters and Turbine operational state)

– $T_{\text{EOL}}$ is the predicted fatigue life,

– $\bar{T}_{\text{EOL}}$ is the scaled fatigue life,

– $T_{\text{EOL,min}}$ and $T_{\text{EOL,max}}$ are the minimum and maximum predicted fatigue life values across all months,

– $\text{Scale}_{\text{min}} = 0.1$ and $\text{Scale}_{\text{max}} = 1$ define the target range for scaled fatigue life suitable for plotting on log-axis.

## 5.1 Feature selection

Feature selection was initially performed using a filter-based method that identifies the top five features for each target variable. These selected features are summarized in Table 3. The numbers in Table 3 refer to the relative importance of the selected features with 1 being most important feature.

For the 3 MW OWT, the results suggest a dominant influence of SCADA parameters across most target variables, with the exception of SS direction, which shows a dependency on mean wave height. In contrast, the 9 MW OWT displays a greater sensitivity to wave-related parameters across all target variables, reflecting the increased susceptibility of larger monopile structures to wave-induced effects as already mentioned by Velarde et al. (2020); Wu et al. (2025). Notably, the SS direction for the 9 MW OWT incorporates all wave parameters except low-frequency wave direction.

In addition to the filter-based approach, RFECV was applied both globally, RFECV-Global, and separately within each turbine operational state, RFECV-Statewise. The results of these RFECV-based feature selection strategies are provided in Appendix B.



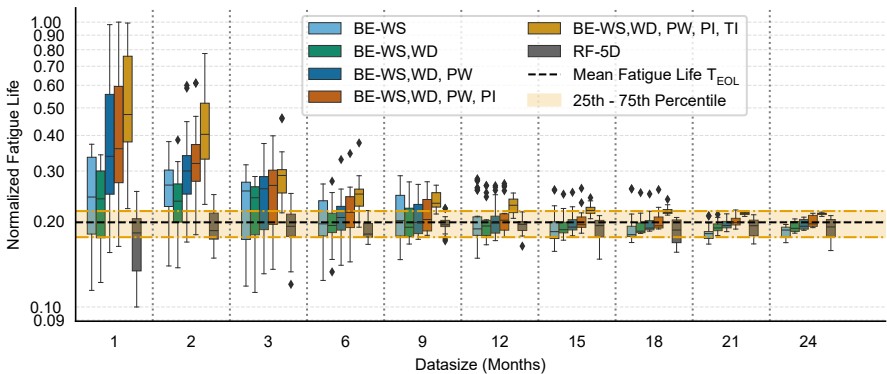

**Figure 6.** Comparison of binning (1D to 5D) and 5D random forest model for a 3MW OWT at 205° single sensor

## 5.2 Single Sensor

The fatigue damage predictions from both binning and RF models vary depending on the size of the monitoring dataset and the
starting point of the data used for training. Figure 6 presents a comparison of fatigue life predictions obtained using binning
(from 1D to 5D) and a 5D RF model for a 3 MW OWT.

The mean fatigue life $T_{EOL}$ denoted by grey dotted line in Fig. 6 represents the true value of fatigue life calculated using
actual measurement data. The $25^{th} - 75^{th}$ percentile between dotted orange lines is calculated by resampling the full monitoring
period with replacement to synthetically generate 100 years of data. Fatigue life is then calculated for each synthetic year to
get a distribution, with mean and percentiles plotted in Fig. 6.

Across all extrapolation models, a general reduction in prediction scatter is observed as the monitoring period increases
confirming the conclusions of Pacheco et al. (2023) stating that the uncertainties in extrapolations decrease with increasing
monitoring period. Notably, the prediction variability in binning approaches decreases with higher binning dimensions (e.g.,
from 1D to 5D). However, higher-dimensional binning tends to over-predict fatigue life, primarily due to the presence of empty
bins. This sensitivity to the bin-filling strategy highlights a limitation of binning with limited data availability.

In contrast, the RF models consistently show lower scatter in predictions for smaller data sizes of 6–9 months. As data
availability increases, predictions from all models begin to converge, with comparable performance across methods for data
sizes beyond 18 months.

Figure 7 shows similar trends for the 9 MW OWT. Higher-dimensional binning again suffers from the presence of empty
bins and heightened sensitivity to the bin-filling strategy, especially at smaller data sizes. The RF model demonstrates superior
performance with significantly lower scatter in predictions for data sizes up to 9 months. While the difference in prediction
variability between RF and binning models diminishes for larger data sizes of 18–21 months, the higher-dimensional binning
methods still tend to overestimate fatigue life.



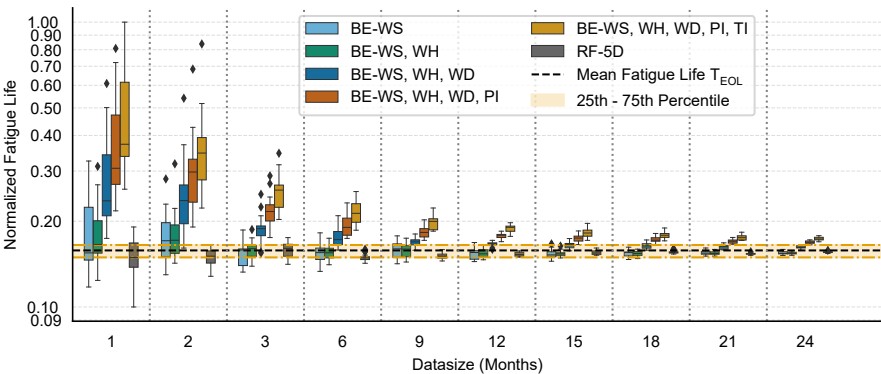

**Figure 7.** Comparison of binning (1D to 5D) and 5D random forest model for a 9MW OWT at 230° single sensor

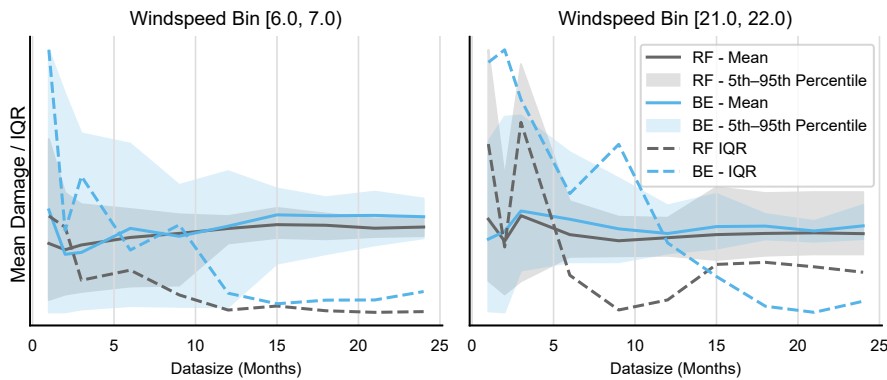

**Figure 8.** Mean and IQR of predicted damages using 1D-binning and 5D-RF model in windpeed bin of [6.0,7.0)-left and [21.0,22.0)-right for a 3MW OWT-single sensor

A comparison of various RF model configurations including feature sets selected using RFECV on global data, RFECV applied separately to each operational state, and models trained using all available features is provided in Fig. C1 and C2 in Appendix C for both 3 MW and 9 MW OWTs. The results suggest that using statewise RFECV and individual models per operational state slightly reduces prediction scatter but introduces additional model complexity.

Figure 8 illustrates the variability in predicted damage across different data sizes and starting points, binned by wind speed intervals, for 1D binning and RF models applied to a 3 MW OWT. As shown in Fig. 8, the mean predicted damage stabilizes after approximately 9–12 months of monitoring data for both the binning and RF models. However, the interquartile range (IQR), representing the spread in predictions, is consistently narrower and converges more rapidly for the RF model, indicating greater robustness to variability in starting point. Notably, higher IQR values persist in the wind speed bin [21.0, 22.0) even after 12 months of data, which can be attributed to the low occurrence frequency of this wind speed range, leading to reduced statistical confidence.




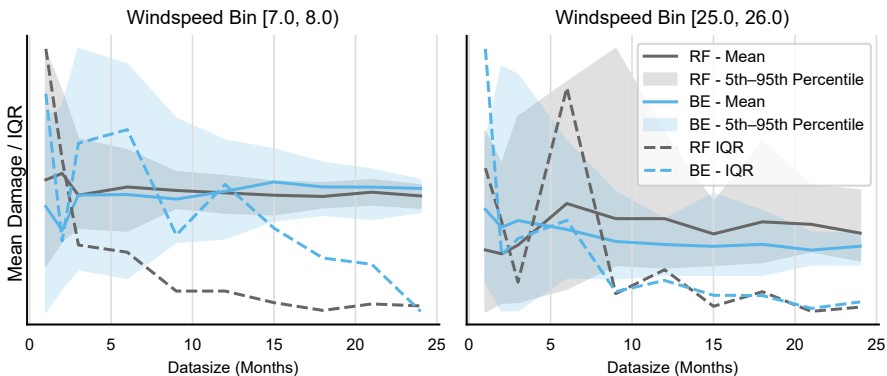

**Figure 9.** Mean and IQR of predicted damages using 1D-binning and 5D-RF model in windpeed bin of [7.0,8.0)-left and [25.0,26.0)-right for a 9MW OWT-single sensor

Similar trends are observed in the damage predictions for the 9 MW OWT, as illustrated in Fig. 9. For the wind speed bin [7.0, 8.0), the IQR of the RF model predictions converges after approximately 9 months of data. In contrast, the IQR for the binning approach continues to decrease gradually with increasing data size, indicating a slower convergence. For the higher wind speed bin [25.0, 26.0), both RF and binning approaches exhibit minor fluctuations in IQR beyond 9–12 months of data. However, the magnitude of these fluctuations remains relatively low.

### 5.3 Fore-Aft direction

Figure 10 presents the comparison of fatigue life predictions in the FA direction for the 3 MW OWT. The RF models exhibit a relatively constant scatter in predictions beyond 9 months of training data. For smaller data sizes of 6–9 months, RF models outperform binning methods showing lower scatter. However, for larger data sizes of greater than 12 months, higher-dimensional binning models, particularly 2D and 3D, show reduced scatter compared to RF models, indicating improved performance in fatigue life estimation.

The variability in RF predictions is further minimized by incorporating additional features, either through RFECV or by training separate RF models for each turbine operational state. The results for these alternative RF configurations are provided in Fig. C3 and C4 in Appendix C.

A similar comparison for the 9 MW OWT is illustrated in Fig. 11. In this case, RF models demonstrate superior performance at smaller data sizes, while all extrapolation methods converge to similar prediction accuracy as the data size increases. Looking at 3 months predictions in Fig. 11, it can be noted that the predictions even for 4D binning are almost within the target region showing lesser influence of empty bins for the 9 MW OWT in FA direction, except in the case of 5D binning, which tends to overestimate fatigue life even when larger data sizes are used.

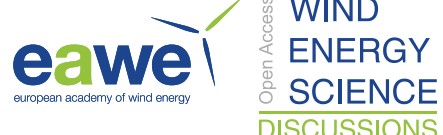

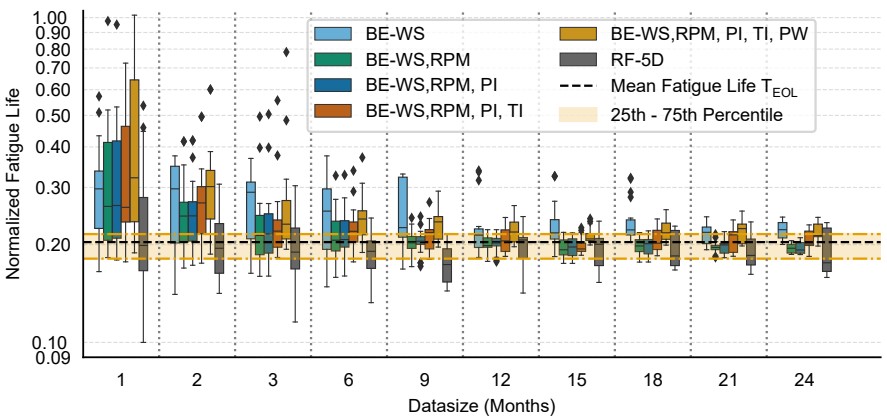

**Figure 10.** Comparison of binning (1D to 5D) and 5D random forest model for a 3MW OWT in Fore-Aft direction

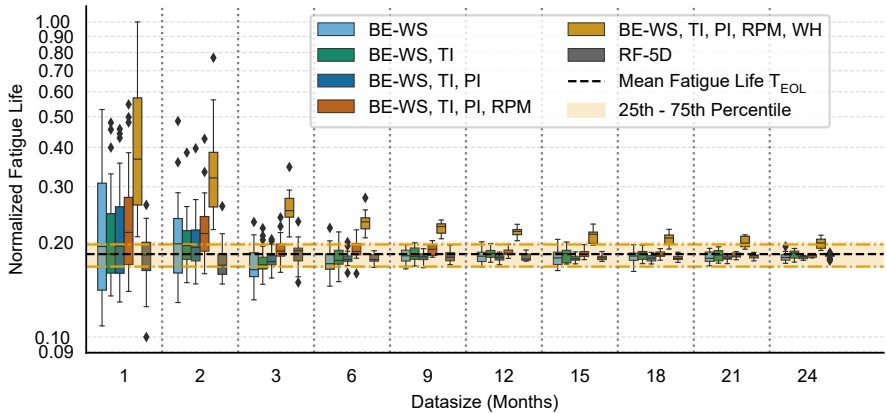

**Figure 11.** Comparison of binning (1D to 5D) and 5D random forest model for a 9MW OWT in Fore-Aft direction

## 5.4 Side-Side direction

Figure 12 illustrates the trends in fatigue life predictions for SS direction in the 3 MW OWT. Similar to the single sensor and FA direction cases, RF models exhibit reduced scatter at smaller data sizes of 6–9 months, and all models—including binning and RF—converge to comparable performance as the data size increases.

For the 9 MW OWT, SS direction predictions show noticeable differences across models, as depicted in Fig. 13. Higher-dimensional binning approaches struggle to provide reliable predictions due to the increased sensitivity to empty bins and
data sparsity. In contrast, RF models maintain consistent and reasonable performance, particularly at smaller data sizes of 6–9 months.





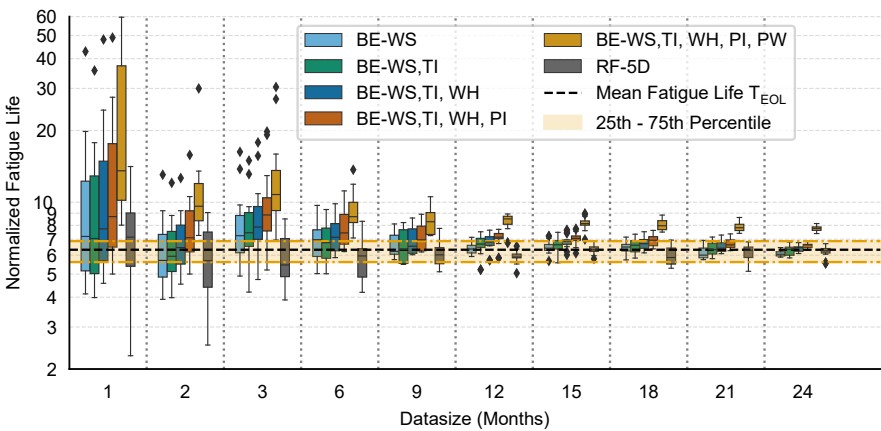

**Figure 12.** Comparison of binning (1D to 5D) and 5D random forest model for a 3MW OWT in Side-Side direction

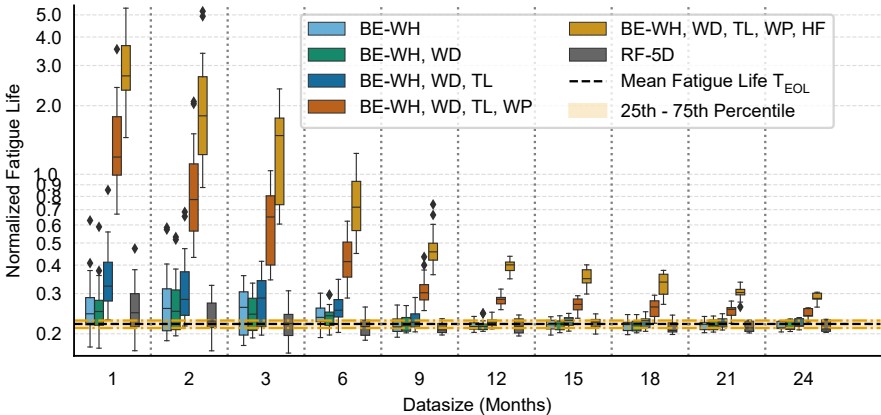

**Figure 13.** Comparison of binning (1D to 5D) and 5D random forest model for a 9MW OWT in Side-Side direction

Notably, no significant improvement in SS direction predictions was observed through the inclusion of additional features, nor by employing RFECV or statewise RF modeling strategies, as summarized in Fig. C5 and C6 in Appendix C.

## 5.5 Prediction performance for different target parameters

The percentage errors over lifetime are defined as given in equation 7, where $T_{\text{EOL, mean}}$ is the mean fatigue life calculated using actual measurement data as described in section 5.2 and $T_{\text{EOL}}$ are the predicted values of fatigue life.

$$\text{Percentage Error over Lifetime} = \left( \frac{T_{\text{EOL}} - T_{\text{EOL, mean}}}{T_{\text{EOL, mean}}} \right) \times 100 \tag{7}$$


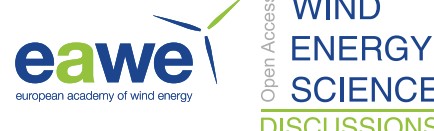

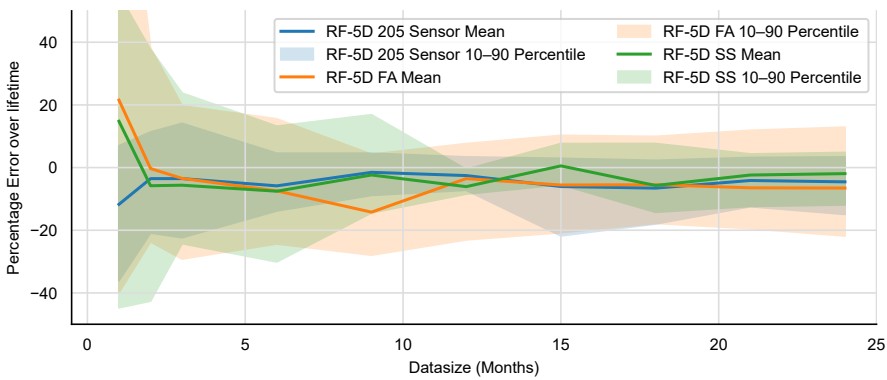

**Figure 14.** Percentage errors in lifetime estimation for random forest models with single sensor, FA and SS directions for 3MW OWT

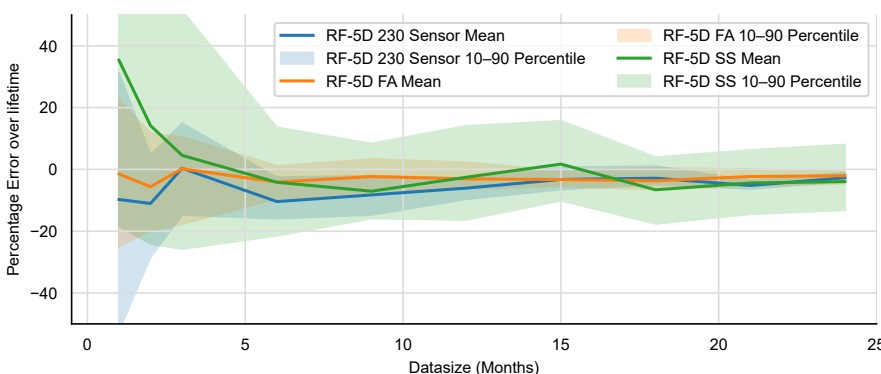

**Figure 15.** Percentage errors in lifetime estimation for random forest models with single sensor, FA and SS directions for 9MW OWT

The percentage errors over lifetime in fatigue life predictions for different target variables, FA, SS and single sensor, associated with varying training data sizes and starting points are presented in Fig. 14 for the 3 MW OWT. For both FA and single
sensor targets, the $10-90^{th}$ percentile of the prediction errors stabilizes after approximately 9 months of training data. In contrast, predictions for the SS direction exhibit continued variability. With 12 months of data, the FA direction yields prediction errors in the range of $[-22\%, +8\%]$, while single sensor predictions remain within a narrower error margin of approximately $\pm 5\%$. SS predictions at 12 months show improved accuracy with errors in the range of $[-8\%, 0\%]$, but the error increases to $[-21\%, +7\%]$ at 15 months before decreasing again with larger data sizes.
Figure 15 shows a similar trend for the 9 MW OWT, where error convergence is observed for all three directions, single sensor, FA, and SS. The overall prediction errors for the 9 MW OWT are lower compared to the 3 MW turbine. Specifically, using 12 months of training data, the RF model predicts lifetime errors for the single sensor target within $[-10\%, -2\%]$, for the FA direction within $[-7\%, +2\%]$, and for the SS direction within approximately $\pm 18\%$.





# 6 Discussions

The features selected through the filter-based feature selection method highlight the greater importance of wave parameters for SS direction in both 3 MW and 9 MW OWTs. Furthermore, the larger 9 MW turbine appears more sensitive to wave-induced fatigue loading compared to the smaller 3 MW turbine.

Analysis of the variation in mean damage values across different wind speed bins for the 3 MW and 9 MW turbines with increasing data sizes indicates that predictions converge after approximately 9–12 months of monitoring data. Additionally, the scatter in predictions is significantly lower for RF models than for binning-based extrapolation. A strain monitoring period of 9-10 months is also recommended in the literature (Hübler et al. (2018); Hübler and Rolfes (2022)) to capture critical fatigue-inducing events, such as high wind speeds and seasonal variability, provided that no significant changes occur in the turbine or its EOCs during this time.

The availability of long-term monitoring data enables the investigation of how the starting point of the monitoring period affects fatigue life estimates. A comprehensive comparison between binning and RF model predictions shows that when accurate fatigue life estimates are required in shorter timeframes of 6-9 months, RF models yield lower statistical uncertainty than binning approaches. RF models are more effective at capturing the nonlinear relationships between SCADA and wave parameters and fatigue damage, particularly when limited data is available. For longer data durations, greater than 15 months, both RF and binning methods provide comparable fatigue life predictions. When long term measurements data is available, binning extrapolation should be preferred as they are more transparent, provide easy control of conservativeness using alternate bin filling strategies and have a stronger link with design as indicated by their similarity to design load cases (DLCs) (IEC (2019); Freudenreich and Argyriadis (2007)).

Increased dimensionality in binning, specially 4D and 5D, results in reduced performance compared to lower-dimensional, 1D and 2D, binning. This is primarily due to the treatment of empty bins. Alternative strategies for empty bin filling, such as using the 90th percentile, the highest neighboring value, or interpolation methods based on surface fitting (Pacheco et al. (2023); Noppe et al. (2020); Hübler and Rolfes (2022)), could enhance the accuracy of high-dimensional binning. In this study, bin sizes are not explicitly optimized. Future work may involve optimizing bin sizes for different data sizes, similar to hyperparameter tuning in RF models, to further improve binning performance.

From a usability perspective, 1D and 2D binning are computationally efficient and easy to implement. However, higher-dimensional binning, 3D and above, becomes increasingly computationally expensive, especially when combined with complex empty-bin filling strategies. On the other hand, RF models, while offering improved prediction accuracy, are more complex and require intensive hyperparameter tuning. More advanced RF pipelines, such as those involving different models for different operational states, can further enhance accuracy but at the cost of increased computational effort and reduced interpretability.

RF models using filter-based feature selection offer a good trade-off between complexity and predictive performance. In practical scenarios where a turbine is instrumented later in its lifetime and a fast, reliable fatigue life estimate is required, RF models can be particularly valuable.



While this study evaluates the statistical uncertainty associated with varying the start of the monitoring period, another layer of uncertainty arises when predictions are made using a fixed data size and start point, especially if the training and extrapolation data differ significantly. Both binning and RF models in this work rely on the mean damage within each bin or the mean prediction, without considering the full distribution. Such uncertainties have been explored in the context of binning by Sadeghi et al. (2024). Future work could explore the use of Quantile Random Forests (Rouholahnejad and Gottschall (2025b)) to quantify this uncertainty. Additionally, Bayesian neural networks may be employed to flag outliers in fatigue life predictions based on differences between training and extrapolation datasets. These advanced uncertainty quantification techniques are beyond the scope of this paper and are proposed as future work.

Finally, it is important to note that the findings of this study are applicable to the wind farms analyzed and, by extension, are likely relevant to the majority of farms located in the North Sea. However, in locations exhibiting differing levels of environmental variability, the applicability of these recommendations may be limited.

## 7   Conclusions

This study concludes that a monitoring period of 9–12 months is required to obtain reliable fatigue life estimates for OWTs, based on case studies using long-term monitoring data from both 3 MW and 9 MW turbines. Feature selection results further demonstrate that the relevance of input parameters depends on the target variable, i.e., single sensor, FA, or SS damage. For the larger 9 MW OWT, wave parameters appear consistently across all target variables, whereas for the smaller 3 MW turbine, only SS fatigue damage includes mean wave height as a key feature. These conclusions specifically apply to offshore wind farms analyzed within the Belgian North Sea, and requires re-evaluation for locations exhibiting varying degrees of environmental variability.

Random forest models exhibit better generalization capability compared to binning-based extrapolation methods, as evidenced by their reduced sensitivity to the starting point of the monitoring period. While 2D binning outperforms 1D binning, higher-dimensional binning, 3D and above, suffers from issues with empty bins, especially when using smaller datasets. Consequently, RF models are more effective for fatigue life prediction when limited data is available, offering lower statistical uncertainty than binning methods.

The predictive advantage of RF models over binning diminishes as the monitoring period increases, beyond 12-15 months, at which point both approaches yield comparable performance. Furthermore, more complex RF models involving additional features or operational state-specific modeling offer only marginal improvements in accuracy.

Future work may enhance the accuracy of binning methods by optimizing bin sizes and systematically evaluating the effects of various empty bin filling strategies. These improvements could help narrow the performance gap between binning and machine learning-based approaches, particularly for high-dimensional binning models.

*Data availability.*   The data used in this paper is proprietary to the industrial partners of this project and cannot be made publicly available





## Appendix A: Effect of different bin filling strategies using 3D binning

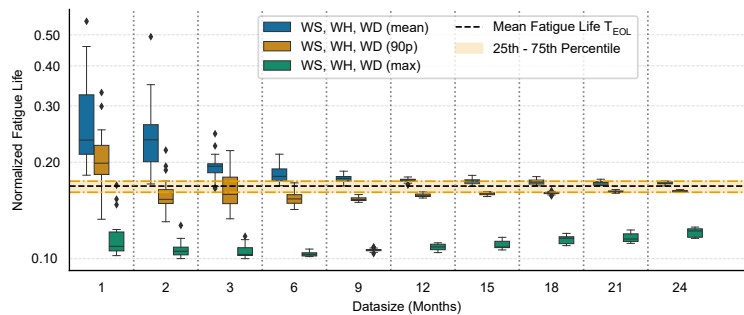

**Figure A1.** Effect of different bin filling strategies on lifetime predictions using 3D binning for a 9MW OWT at 230° single sensor

## Appendix B: Feature selection using RFECV-Global and Statewise

**B1 RFECV-Global features**

RFECV was applied to 12 months of data to identify the most relevant features for fatigue life prediction. Operational states were incorporated into the feature space using one-hot encoding. The selected features for both the 3 MW and 9 MW offshore wind turbines are given in Table B1.

**Table B1.** Selected features for different target variable for 3MW and 9MW OWT using RFECV on 12 months of data

|  | 3MW OWT | | | 9MW OWT | | |
|---|---|---|---|---|---|---|
| **Variable** | Single sensor 205° | Fore-Aft | Side-Side | Single sensor 230° | Fore-Aft | Side-Side |
| Windspeed | ✔ | ✔ | ✔ |  | ✔ | ✔ |
| Yaw Angle | ✔ |  | ✔ | ✔ | ✔ | ✔ |
| Pitch Angle | ✔ | ✔ | ✔ | ✔ | ✔ |  |
| Power | ✔ | ✔ | ✔ | ✔ | ✔ |  |
| Rotational Speed | ✔ | ✔ | ✔ | ✔ | ✔ |  |
| Turbulence Intensity | ✔ | ✔ | ✔ | ✔ | ✔ | ✔ |
| Operational State* | ✔ | ✔ | ✔ | ✔ | ✔ | ✔ |
| Mean Wave Height | ✔ | ✔ | ✔ | ✔ | ✔ | ✔ |
| Mean Wave Period |  |  | ✔ |  | ✔ | ✔ |
| High Frequent Wave Direction | ✔ |  | ✔ |  |  | ✔ |
| Low Frequent Wave Direction |  |  | ✔ |  |  | ✔ |
| Tidal level |  |  | ✔ |  |  | ✔ |

* Operational state is deliberately included in all models



## B2 RFECV-Statewise features

RFECV was applied to 12 months of data by first filtering the dataset into individual operational states and performing RFECV separately on each subset. The resulting RFECV-Statewise feature sets for both the 3 MW and 9 MW offshore wind turbines are given in Table B2 and B3.

**Table B2.** Operational state-wise selected features using RFECV-Statewise on 12 months of data for 3MW

| Operational State → <br> Parameters ↓ | Single sensor 205° | | | | Fore-Aft | | | | Side-Side | | | |
|---|---|---|---|---|---|---|---|---|---|---|---|---|
| | 1 | 2 | 3 | 4 | 1 | 2 | 3 | 4 | 1 | 2 | 3 | 4 |
| Windspeed | ✔ | ✔ | ✔ | ✔ | ✔ | ✔ | ✔ | ✔ | ✔ | ✔ | ✔ | ✔ |
| Yaw Angle | ✔ | ✔ | ✔ | ✔ | ✔ | ✔ | ✔ | ✔ | ✔ | ✔ | ✔ | |
| Pitch Angle | ✔ | ✔ | | ✔ | ✔ | ✔ | | ✔ | ✔ | | | |
| Power | ✔ | | ✔ | ✔ | ✔ | ✔ | ✔ | ✔ | ✔ | ✔ | ✔ | |
| Rotational Speed | ✔ | | ✔ | ✔ | ✔ | ✔ | ✔ | ✔ | | ✔ | ✔ | |
| Turbulence Intensity | ✔ | | ✔ | ✔ | ✔ | ✔ | ✔ | ✔ | ✔ | ✔ | ✔ | ✔ |
| Operational State* | ✔ | ✔ | ✔ | ✔ | ✔ | ✔ | ✔ | ✔ | ✔ | ✔ | ✔ | ✔ |
| Mean Wave Height | ✔ | ✔ | ✔ | ✔ | ✔ | ✔ | ✔ | ✔ | ✔ | ✔ | ✔ | ✔ |
| Mean Wave Period | | ✔ | ✔ | ✔ | | ✔ | ✔ | ✔ | ✔ | ✔ | ✔ | |
| High Frequent Wave Direction | ✔ | ✔ | ✔ | ✔ | | ✔ | ✔ | ✔ | | ✔ | ✔ | ✔ |
| Low Frequent Wave Direction | ✔ | ✔ | ✔ | ✔ | ✔ | ✔ | ✔ | ✔ | ✔ | ✔ | ✔ | |
| Tidal level | | ✔ | ✔ | ✔ | | ✔ | ✔ | ✔ | ✔ | ✔ | ✔ | ✔ |

1: Nominal, 2: Idling, 3: Highwind, 4: Invalid

## Appendix C: Comparison of different RF configurations

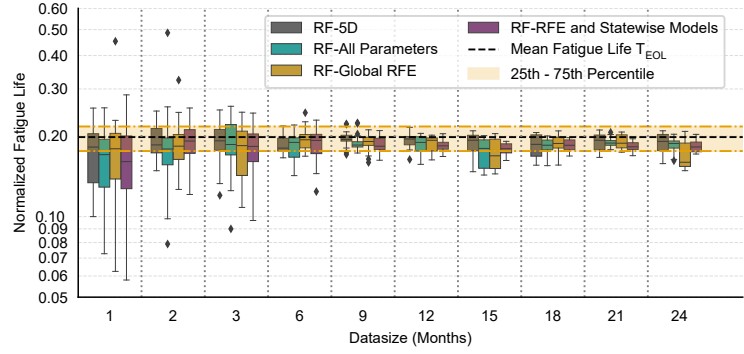

**Figure C1.** Comparison of different RF model configurations for a 3MW OWT at 205° single sensor



**Table B3.** Operational state-wise selected features using RFECV-Statewise on 12 months of data for 9MW

| Parameters ↓ \ Operational State → | Single sensor 230° | | | | Fore-Aft | | | | Side-Side | | | |
|---|---|---|---|---|---|---|---|---|---|---|---|---|
| | 1 | 2 | 3 | 4 | 1 | 2 | 3 | 4 | 1 | 2 | 3 | 4 |
| Windspeed | ✔ | ✔ | ✔ | ✔ | ✔ | ✔ | ✔ | ✔ | ✔ | ✔ | | ✔ |
| Yaw Angle | ✔ | | ✔ | | | ✔ | | | ✔ | ✔ | ✔ | |
| Pitch Angle | ✔ | | ✔ | ✔ | ✔ | | ✔ | | | ✔ | | ✔ |
| Power | ✔ | | ✔ | ✔ | | ✔ | ✔ | ✔ | ✔ | | | ✔ |
| Rotational Speed | ✔ | | ✔ | ✔ | ✔ | ✔ | ✔ | ✔ | ✔ | | | ✔ |
| Turbulence Intensity | ✔ | | ✔ | ✔ | | ✔ | ✔ | ✔ | ✔ | ✔ | ✔ | ✔ |
| Operational State* | ✔ | ✔ | ✔ | ✔ | ✔ | ✔ | ✔ | ✔ | ✔ | ✔ | ✔ | ✔ |
| Mean Wave Height | ✔ | ✔ | ✔ | ✔ | ✔ | ✔ | ✔ | ✔ | ✔ | ✔ | ✔ | ✔ |
| Mean Wave Period | ✔ | | ✔ | ✔ | | ✔ | ✔ | ✔ | ✔ | ✔ | ✔ | ✔ |
| High Frequent Wave Direction | ✔ | | ✔ | | | ✔ | ✔ | ✔ | ✔ | ✔ | ✔ | ✔ |
| Low Frequent Wave Direction | | | ✔ | ✔ | | ✔ | ✔ | ✔ | ✔ | ✔ | ✔ | ✔ |
| Tidal level | ✔ | | ✔ | ✔ | | ✔ | ✔ | ✔ | ✔ | ✔ | ✔ | ✔ |

1: Nominal, 2: Idling, 3: Abnormal, 4: Invalid

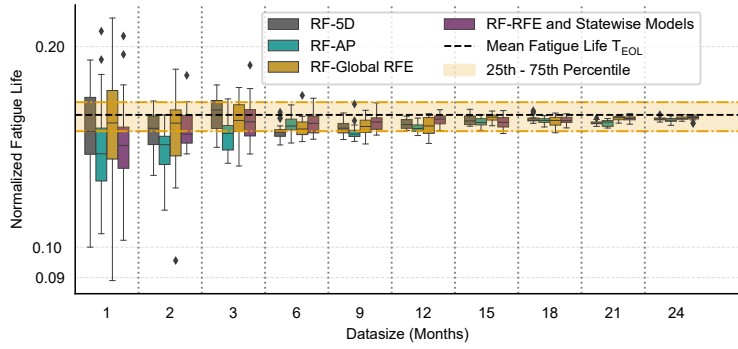

**Figure C2.** Comparison of different RF model configurations for a 9MW OWT at 230° single sensor

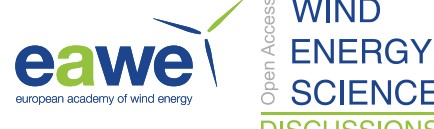

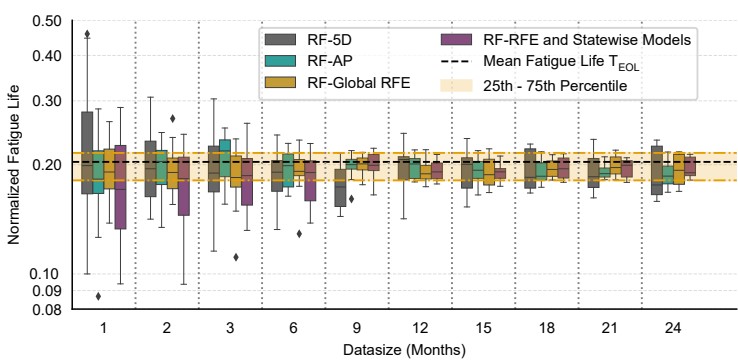

**Figure C3.** Comparison of different RF model configurations for a 3MW OWT in FA direction

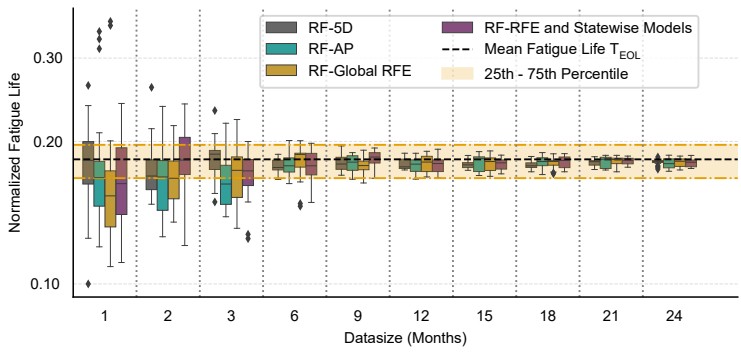

**Figure C4.** Comparison of different RF model configurations for a 9MW OWT in FA direction

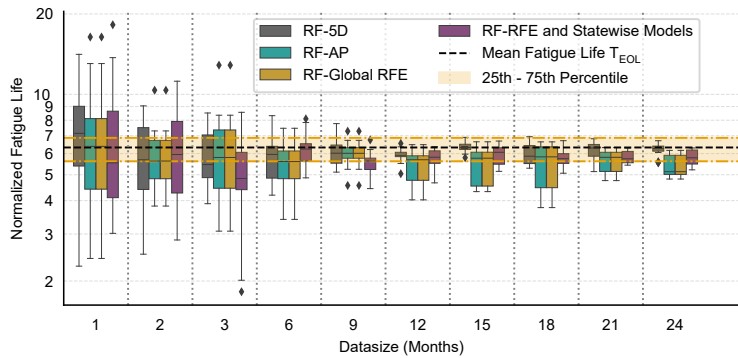

**Figure C5.** Comparison of different RF model configurations for a 3MW OWT in SS direction

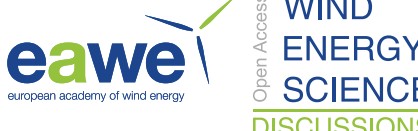

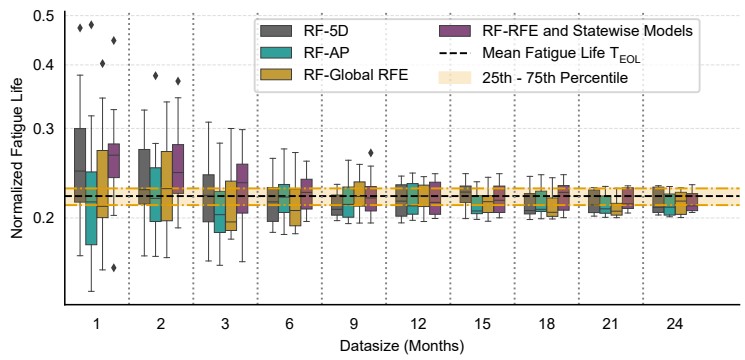

**Figure C6.** Comparison of different RF model configurations for a 9MW OWT in SS direction

*Author contributions.* AM: conceptualizing, analyzing, interpretation, writing. WW: conceptualizing, validation, interpretation, supervision.
NS: interpretation, validation. CD: supervision, data collection, validation. All authors contributed to the review and editing.

*Competing interests.* The authors have no competing interests.

*Acknowledgements.* We acknowledge Parkwind and Norther for their permission to use the monitoring data. We acknowledge the support of
VLAIO through De Blauwe Cluster cSBO FIRMEST project. We acknowledge WILLOW project, funded by the European Union with GA
No. 101122184. Views and opinions expressed are however those of the author(s) only and do not necessarily reflect those of the European
Union. Neither the European Union nor the granting authority can be held responsible for them.

During the preparation of this work, the author(s) used ChatGPT5 to improve the text. After using this tool/service, the author(s) reviewed
and edited the content as needed and take(s) full responsibility for the content of the publication.





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
