# Peer review of "A machine learning based approach for better prediction of fatigue life of offshore wind turbine foundations using smaller datasizes"

_Wind Energy Science, 2025_

## Referee Comment (RC1)

**"A machine learning based approach for better prediction of fatigue life of offshore wind turbine foundations using smaller datasizes" (https://doi.org/10.5194/wes-2025-173)**

In this work, a new machine learning-based approach for strain measurement-based fatigue lifetime assessments of offshore wind turbine substructures is presented. The approach is based on a random forest model for the temporal extrapolation of the strain measurements. The approach is compared to state-of-theart binning approaches and is validated using unique five-year data sets for a 3MW and a 9MW turbine.

The overall topic is very relevant, as many offshore wind turbines start to reach the end of their design lifetime and lifetime extensions based on strain measurements can be very valuable. The idea of using machine learning-based approaches for the temporal extrapolation is not new, although random forest models have not been used so far. However, although no very innovative approach is proposed, the work is definitely of interest to the research community, as the extend of the validation campaign is unique and allows long-term validation (or at least validation using data of five years). The results are interesting and confirm previous results but using a more reliable and broader database.

The paper is well written and nicely structured but would benefit from several minor clarifications and additions.

**Comments:**

- 1) In your work, you use SCADA and wave data for the extrapolation. Surely, SCADA should always be available. However, wave data might not always be available or at least not for the precise location (or close to it). Perhaps, you can briefly discuss how far away from the turbine the wave measurements can be conducted to be still useful.
- 2) In the abstract, you write "However, for longer datasets, greater than 12 months, the performance advantage of RF model over binning methods becomes less pronounced." Based on the presented results and your discussion, I would conclude that "for longer datasets, greater than 12 months, the performance advantage of RF model over binning methods disappears."
- 3) L. 105-113: Perhaps, you can make even more clear, what is actually missing and what has already been done. For example, the 4th bullet point has been investigated before, just as the 1st point (but only for a single turbine)
- 4) In Fig. 1, it is not clear what the green area/arrow between Tmoe and Tcurr is. In Fig. 4, this becomes clear. Nevertheless, perhaps, you can already explain it here.
- 5) L. 139/140: You write "During the monitoring period  $\Delta T_m = T_{moe} T_{moi}$ , extrapolation models are trained using measured strain and corresponding SCADA/ wave data." This sounds as if the entire data is used for training. However, if I am not mistaken, you also use the data for validation and testing. Hence, perhaps, you can reformulate it.
- 6) Fig. 2: It is not stated what we see on the axis (I assume that it is the probability). Furthermore, it is not clear why the wind rose of the 9MW turbine is bigger than the one of the 3MW turbine.
- 7) Section 3.1: You use quite a lot of factors/parameters (e.g., SCF, MSF, m, k, linearly or logarithmic spaced bins etc.). Although the precise values are not so relevant for this study, it would be nice if you could state them or refer to some publications where the reader can look them up.
- 8) L. 199: You state that using FA is always conservative. It this actually the case? For the 9MW turbine, depending on the site, it might be the case that SS is more damaging due to the reduced aerodynamic damping. Perhaps, you can comment on that for your site+turbine.
- 9) L. 209: The state "curtailed" mentioned here is not mentioned in Table 1. Furthermore, in l. 211, you state that "abnormal = curtailed" for the 9MW turbine. Perhaps, you can clarify this.

- 10) L. 252: You state "A filter-based method is applied using the built-in feature importance metric". Could you provide some more information about the feature importance metric or at least refer to a publication etc. where it is explained in more detail.
- 11) L. 253: You state that you use a 12-month period. Perhaps, you can add which one.
- 12) L. 259: "A recursive feature elimination with cross-validation is applied": Again, could you provide some more information or refer to a relevant publication?
- 13) L. 281: Not using optimized bin sizes is totally fine. Still, it would be nice to know what bin sizes you used.
- 14) L. 323: The SP is selected randomly. Is it allowed that the SP is so late that the measurement period goes beyond the end of the overall period or is the latest possible SP  $T_{curr} \Delta T_m$ ? In case, you allow very late SP, how do you do this, e.g., continue the measurement period at the beginning, i.e., at  $T_o$ ? If you do not allow so late SP, you should briefly discuss this, as it adds a slight bias in the data selection (data close to  $T_o$  is less likely to be chosen).
- 15) L. 330/Fig. 5: Is it correct that you generate just a single synthetic 20-year period by resampling from the 5-year period. And then, you reduce it to one year again by taking the mean yearly damage. I do not really understand why you generate the 20-year period. Surely, this is useful if you want to analyze the scatter between different 20-year periods. However, as you only have a single one, you could just take the five-year period and determine the mean yearly damage from that or am I mistaken?
- 16) L. 354: What do you mean by "all months". I assume, all 20 SP + 10  $\Delta T_m$  + all approaches (binning 1D, 2D, ... RF)? At least this is what it looks like in Fig. 6.
- 17) L. 374: You describe that you generate 100 random years by resampling from the 5-year period. For each year, the fatigue life is calculated. In my opinion, this leads to too high uncertainty, as the fatigue life calculation is based on a single year. In my opinion, you should use 100 random 20-year periods to be consistent with the one 20-year period used for the extrapolation methods.
- 18) L. 381: You state "As data availability increases, predictions from all models begin to converge, with comparable performance across methods for data sizes beyond 18 months." In my opinion, for large data sizes and this case (3MW, single sensor), RF is outperformed by the binning approaches, as the uncertainty of RF is higher. It is totally fine that RF is outperformed by the binning approach for data sizes beyond 18 months. However, we should not present the ML approach in an overly positive light. For the 9MW turbine (I. 387), I agree with your statement that the difference diminishes.
- 19) L. 393-399: In my opinion, a discussion of the higher IQR of RF compared to the binning approach for the wind speed bin 21-22 m/s is missing. This higher IQR is probably also the reason why RF is outperformed by the binning approach in Fig. 6 (see comment 18)
- 20) L. 404: You state: "However, the magnitude of these fluctuations remains relatively low." Fig. 9 has no scale on the axis. Hence, it is hard to judge how high the fluctuations are. Nonetheless, if you look at the 5-95 percentile, it looks as if RF is not well performing for this wind speed bin (Fig. 9b, 25-26m/s).
- 21) L. 425: Again, I think that you highlight the benefits of RF too much. Not only "RF models maintain consistent and reasonable performance, particularly at smaller data sizes of 6–9 months" but 1D and 2D bins as well.
- 22) L. 435: "For both FA and single sensor targets, the 10–90th percentile of the prediction errors stabilizes after approximately 9 months of training data. In contrast, predictions for the SS direction exhibit continued variability." Where do I see this difference in stabilisation? I assume that the low value at 12 months (preventing a good stabilisation) is just a statistical artifact/outlier
- 23) L. 449: The statement "Additionally, the scatter in predictions is significantly lower for RF models than for binning-based extrapolation." is only true for short measurement periods. For longer ones, the scatter for RF models is sometimes even higher. Again, I do not say that RF has no clear advantages, but we should not present it in an overly positive light
- 24) L. 454: You state that effects of the "starting point of the monitoring period" can by analysed. This is true but I would be careful with the statement, as, in my opinion, such an analysis has not be done in this work.

- You analysed the influence of the measurement length but did not check whether a starting point in winter/summer makes a difference.
- 25) L. 497: You state that "2D binning outperforms 1D binning". This has not been discussed anywhere in the paper and is not clearly visible in the data. I recommend to either discuss it somewhere else using the available figures or to remove this statement in the "conclusions"

**Typos etc.:**

- 26) L. 122: Remove space after "Objective section".
- 27) L. 140: Remove space after "SCADA/".
- 28) Be consistent when writing "10-minute data" etc. Sometimes you write it with and sometime without hyphen.
- 29) L. 180: Remove space before (Hubler and Rolfes (2022)).
- 30) Table 1: I would use "s" and not "sec" and "o" and not "Deg"
- 31) L. 342: "the model comparison" and not "the models comparison"
- 32) L. 344: LFFDfactor and not only LFFD
- 33) L. 350: Be consistent when using multiplication signs ( $\cdot$  or preferable  $\times$ )
- 34) Table 3: Be consistent with the spelling of "windspeed/wind speed"
- 35) Table 3: "Turbine" should not be capitalized.
- 36) L. 431: Be consistent with the capitalisation of "Section" etc. and their abbreviations (e.g., Fig. vs Figure)
- 37) Table B2 and B3: Remove the \* from "Operational State\*" or state what it means.

---

## Author Comment (AC1)

Dear Reviewer,

We sincerely thank you for the constructive and detailed feedback provided in your review. We greatly appreciate your thorough assessment of our work and the valuable comments that have helped us strengthen the clarity, accuracy, and overall quality of the manuscript.

We have carefully addressed each comment and have revised the manuscript accordingly, providing additional explanations, clarifications, and references where necessary.

We genuinely appreciate the time and effort you have devoted to reviewing our work.

In our response:

- The reviewer comments are in Black and
- Author Replies are in Red, and
- The changes made to the manuscript are marked in Blue

**"A machine learning based approach for better prediction of fatigue life of offshore wind turbine foundations using smaller datasizes" (https://doi.org/10.5194/wes-2025-173)**

In this work, a new machine learning-based approach for strain measurement-based fatigue lifetime assessments of offshore wind turbine substructures is presented. The approach is based on a random forest model for the temporal extrapolation of the strain measurements. The approach is compared to state-of-the art binning approaches and is validated using unique five-year data sets for a 3MW and a 9MW turbine.

The overall topic is very relevant, as many offshore wind turbines start to reach the end of their design lifetime and lifetime extensions based on strain measurements can be very valuable. The idea of using machine learning based approaches for the temporal extrapolation is not new, although random forest models have not been used so far. However, although no very innovative approach is proposed, the work is definitely of interest to the research community, as the extend of the validation campaign is unique and allows long-term validation (or at least validation using data of five years). The results are interesting and confirm previous results but using a more reliable and broader database.

The paper is well written and nicely structured but would benefit from several minor clarifications and additions.

**Comments:**

1) In your work, you use SCADA and wave data for the extrapolation. Surely, SCADA should always be available. However, wave data might not always be available or at least not for the precise location (or close to it). Perhaps, you can briefly discuss how far away from the turbine the wave measurements can be conducted to be still useful.

We thank the reviewer for this comment. We agree that while SCADA data are available at each turbine, wave measurements are not always acquired directly at the turbine location and may come from nearby buoys or coastal stations.

In this study, tidal level data used for the 9 MW turbine were obtained from the Ostend station, located approximately 29 km away. Because tidal variations evolve slowly over large distances, the tidal levels at Ostend agree very well with those measured closer to the 9MW OWT location. However, the latter were not available for the full monitoring period.

[Figure]

*Figure 1: Comparison of general trends of Tide measurements at 9MW OWT location and Ostend station*

For wave-related parameters, measurements were taken from the Thorntonbank South buoy, located 18.5 km from the 3 MW turbine and 7.8 km from the 9 MW turbine. Although some differences in wave amplitude exist between the buoy and local measurements, the overall trends are consistent. Since the extrapolation models primarily rely on capturing these broader trends rather than values, the buoy data remain suitable for the extrapolation task. It does raise some concern when as-designed wave conditions, rather than wave distributions taken from the measurement station, would be used for the extrapolation. In such a scenario, the transformation needs to be properly checked.

[Figure]

*Figure 2: Comparison of wave height measurements (top) and wave period measaurements (bottom) from 9MW OWT location and Thorntonbank buoy*

More generally, the acceptable distance between wave measurements and a turbine depends on local bathymetry, wave climate, and directional spreading. Nonetheless, we acknowledge that in sites with strong spatial gradients, closer wave measurements would be necessary.

In response, we have added the following description in the manuscript:

"The distance between the turbine and the wave measurement buoy leads to some differences in wave parameters compared with measurements taken closer to the 9~MW turbine, although the overall temporal trends remain consistent. However, the acceptable distance between wave measurement station and a turbine depends on local bathymetry and wave climate, and closer measurements may be required in areas with strong spatial gradients."

2) In the abstract, you write "However, for longer datasets, greater than 12 months, the performance advantage of RF model over binning methods **becomes less pronounced**." Based on the presented results and your discussion, I would conclude that "for longer datasets, greater than 12 months, the performance advantage of RF model over binning methods **disappears**."

We thank the reviewer for this comment. We agree that for certain cases such as for 3MW OWT in the single sensor and Fore-Aft (FA) direction, the performance advantage of the RF model over the binning approach disappears for datasets greater than 18 months. However, this trend does not hold generally across all cases in our study.

For other configurations (e.g. for 9MW OWT in single sensor, FA and Side-Side (SS) direction), the RF model continues to exhibit consistently lower scatter than the binning

based extrapolation even for datasets longer than 12 months. In these cases, although the performance gap narrows significantly with larger datasets, it does not fully disappear.

We acknowledge that binning performance could potentially be improved through optimized bin size definitions or alternate bin-filling strategies. However, such optimizations were outside the scope of the present work.

In response, we have added the following sentence in the abstract for clarification:

"For 3MW OWT with datasets greater than 18 months, RF model is outperformed by binning methods."

3) L. 105-113: Perhaps, you can make even more clear, what is actually missing and what has already been done. For example, the 4th bullet point has been investigated before, just as the 1st point (but only for a single turbine)

We thank the reviewer for this comment. In response, we have added the following text to more clearly distinguish between what has already been investigated in the literature and what remains missing:

*"While previous studies have compared binning approaches with machine learning models such as ANNs and GPR (Hübler and Rolfes (2022); de N Santos et al. (2023)), these comparisons were generally performed for a single turbine and often using relatively limited datasets. Similarly, earlier work has examined the sensitivity of data-driven fatigue extrapolation to dataset length and measurement start time (Hübler and Rolfes (2022)), but again typically for a single turbine. Consequently, a systematic evaluation of Random Forest models across multiple turbine sizes using extensive monitoring data, together with a multi-directional assessment of model performance and parameter relevance remains missing."*

This addition explicitly acknowledges prior work on topics related to the first and fourth bullet points, while clarifying the broader multi-turbine, long-term comparison that the present study contributes.

4) In Fig. 1, it is not clear what the green area/arrow between $T_{mo_e}$ and $T_{curr}$ is. In Fig. 4, this becomes clear. Nevertheless, perhaps, you can already explain it here.

*We thank the reviewer for this comment. In response, we have added a clarification in the paper to explain the meaning of the green shaded region/arrow between $T_{mo_e}$ and $T_{curr}$. The revised text now reads:*

*"The shaded green region between $T_{mo_e}$ and $T_{curr}$ indicates that the monitoring campaign may end before the current date. This represents the period for which no strain measurements are available. In cases where monitoring is still ongoing $T_{mo_e}$ and $T_{curr}$ coincide."*

5) L. 139/140: You write "During the monitoring period $\Delta T_m = T_{mo_e} - T_{mo_i}$ , extrapolation models are trained using measured strain and corresponding SCADA/ wave data." This sounds as if the entire data is used for training. However, if I am not mistaken, you also use the data for validation and testing. Hence, perhaps, you can reformulate it.

We thank the reviewer for this helpful observation. The original wording may indeed give the impression that the entire monitoring dataset is used exclusively for model training, which is not the case.

To clarify this, we have revised the sentence as follows:

*"During the monitoring period $\Delta T_m = T_{mo_e} - T_{mo_i}$ , extrapolation models are trained/ tested and validated using measured strain and corresponding SCADA/ wave data."*

6) Fig. 2: It is not stated what we see on the axis (I assume that it is the probability). Furthermore, it is not clear why the wind rose of the 9MW turbine is bigger than the one of the 3MW turbine.

We thank the reviewer for this comment. We have updated the description of Figure 2 to clarify the axis meaning and the difference in windrose size.

*"Figure 2 presents the windrose diagrams for the 3 MW and 9 MW OWT, illustrating the probability distribution of wind occurrence across different directions. Figure 2 also shows the sensor locations positioned around the circumference, together with the dominant wind direction of 230°. The difference in windrose size reflects the actual relative monopile diameters: the smaller windrose on Fig. 2 (left) corresponds to a smaller diameter monopile for the 3 MW OWT, while the larger windrose on Fig. 2 (right) represents the larger diameter monopile of the 9 MW OWT. The sensor closest to the dominant wind direction is located at 205° for the 3 MW OWT and at 230° for the 9 MW OWT."*

7) Section 3.1: You use quite a lot of factors/parameters (e.g., SCF, MSF, m, k). Although the precise values are not so relevant for this study, it would be nice if you could state them or refer to some publications where the reader can look them up.

We thank the reviewer for this helpful comment. In response, we have added references to the relevant standards that define these parameters and clarified the specific S-N curve used in the study. The precise values of the parameters are not central to our paper, however, the added references allow interested readers to locate the detailed definitions.

"Additional correction factors include a stress concentration factor (SCF) as defined in DNV-RP-C203 (2024) for various fatigue details, a material safety factor (MSF) as specified in relevant design guidelines such as IEC-61400-1 (2019), and a thickness correction factor following the recommendations in DNV-RP-C203 (2024) to account for the size effects."

"The slope (m) and intercept ($\log(\bar{a})$) defining the S-N curve depend on the fatigue detail, environmental conditions, and material properties, and are provided in DNV-RP-C203 (2024) guidelines. In this study, the bilinear DNV D-A curve for a D-detail (circumferential butt weld made from both sides) is adopted, with slopes of m = [3, 5] and intercepts of $\log(\bar{a})$ = [12.164, 15.606], with a transition at $10^7$ cycles."

8) L. 199: You state that using FA is always conservative. It this actually the case? For the 9MW turbine, depending on the site, it might be the case that SS is more damaging due to the reduced aerodynamic damping. Perhaps, you can comment on that for your site+turbine.

We thank the reviewer for this comment. We agree that for some load cases side-side direction might be more damaging specially for 9MW OWT than the fore-aft.

We have clarified the statement in manuscript as follows:

"**Fore-Aft (FA) Damage:** It is assumed that the turbine faces fore-aft damage conditions throughout its lifetime. This method likely underestimates fatigue life as compared to a single sensor, since FA damage state is not constantly maintained over time."

In terms of lifetime, it is conservative to assume FA all the time rather than any fixed heading (single sensor). We cannot tell the same thing for SS damage scenario, though.

9) L. 209: The state "curtailed" mentioned here is not mentioned in Table 1. Furthermore, in l. 211, you state that "abnormal = curtailed" for the 9MW turbine. Perhaps, you can clarify this.

We thank the reviewer for this observation. The curtailed operation of the 9MW OWT is labelled as "abnormal" in the data. We have updated the manuscript to replace abnormal with curtailed for clarity and consistency.

"For the 9MW OWT, an additional curtailed operational state is identified, associated with curtailed power output, defined as intentional operation of the turbine at reduced power output to satisfy grid requirements."

10) L. 252: You state "A filter-based method is applied using the built-in feature importance metric". Could you provide some more information about the feature importance metric or at least refer to a publication etc. where it is explained in more detail.

We thank the reviewer for this helpful comment. In response, we have added a brief explanation of the feature-importance metric and a reference to the original Random Forest publication:

"A filter-based method is applied using the built-in feature importance metric from a Random Forest model, which ranks input variables based on their contribution to reducing node impurity across all trees (see Breiman (2001, 2002) for more details)."

This explanation clarifies that the feature importance is derived from the mean decrease in impurity as originally defined in Breiman's Random Forest framework.

11) L. 253: You state that you use a 12-month period. Perhaps, you can add which one.

We thank the reviewer for this comment. To clarify, we have added the following sentence:

"This model is trained using the first 12 months of monitoring data."

This makes explicit that the initial 12-month segment of the monitoring period is used for feature-importance analysis.

12) L. 259: "A recursive feature elimination with cross-validation is applied": Again, could you provide some more information or refer to a relevant publication?

We thank the reviewer for this comment. We have now added a reference to the original work introducing recursive feature elimination:

"A recursive feature elimination with cross-validation (see Guyon et al. (2002) for details on RFECV methodology) is applied."

13) L. 281: Not using optimized bin sizes is totally fine. Still, it would be nice to know what bin sizes you used.

We thank the reviewer for this comment. We have added the bin sizes in the Appendix and have referred them in the main text:

"Bin sizes for SCADA and wave parameters used in this paper are given in Appendix D."

Appendix D: Bin sizes

*Table D1: Bin sizes for SCADA and wave parameters*

| Variable | Windspeed | Yaw Angle | Pitch Angle | Power | Rotational Speed | Turbulence Intensity | Mean Wave Height | Mean Wave Period | High Frequent Wave Direction | Low Frequent Wave Direction | Tidal level |
|---|---|---|---|---|---|---|---|---|---|---|---|
| Units | m/sec | Degree | Degree | kW | rpm | % | cm | sec | Degree | Degree | cm |
| Bin Size | 1 | 10 | 5 | 250 | 0.5 | 5 | 10 | 0.2 | 10 | 10 | 10 |

14) L. 323: The SP is selected randomly. Is it allowed that the SP is so late that the measurement period goes beyond the end of the overall period or is the latest possible SP $T_{curr} - \Delta T_m$? In case, you allow very late SP, how do you do this, e.g., continue the measurement period at the beginning, i.e., at $T_o$? If you do not allow so late SP, you should briefly discuss this, as it adds a slight bias in the data selection (data close to $T_o$ is less likely to be chosen).

We thank the reviewer for this insightful comment. In our implementation, the random selection of the start point (SP) is explicitly constrained such that the monitoring period of length $\Delta T_m$ never extends beyond the available dataset. Thus, the latest allowable SP is $T_{curr} - \Delta T_m$, and we do not permit monitoring periods that would extend past $T_{curr}$ or wrap around to the beginning of the dataset at $T_o$. We acknowledge that this constraint implies a slight underrepresentation of data near the start of the monitoring period ($T_o$), since inclusion of these points requires selecting a SP very close to $T_o$. However, given the long duration of the dataset, this minor bias does not materially affect the results. To clarify, we have added the following description in the paper:

"To ensure that each monitoring period is fully contained within the available dataset, the start point (SP) is restricted such that the monitoring period $\Delta T_m$ does not extend beyond $T_{curr}$ thus, the latest allowable SP is $T_{curr} - \Delta T_m$."

15) L. 330/Fig. 5: Is it correct that you generate just a single synthetic 20-year period by resampling from the 5-year period. And then, you reduce it to one year again by taking the mean yearly damage. I do not really understand why you generate the 20-year period. Surely, this is useful if you want to analyze the scatter between different 20-year periods.

However, as you only have a single one, you could just take the five year period and determine the mean yearly damage from that or am I mistaken?

We thank the reviewer for this helpful comment. The original purpose of generating a synthetic 20-year period through resampling was to enable the assessment of year-to-year damage variability over a full design-life duration. Although the algorithm internally computes yearly damages across this synthetic period, the present paper uses only the mean yearly damage for calculating the end-of-life $T_{EOL}$. As the reviewer correctly notes, because $T_{EOL}$ depends only on the mean yearly damage, using the entire 5-year dataset directly (without constructing a synthetic 20-year period) would lead to essentially the similar mean yearly damage value. Therefore, the resampling step does not influence the conclusions of this work.

To clarify this in the manuscript, we have added the following sentence:

"The resampling of the 5-year monitoring period into a synthetic 20-year sequence is primarily useful when analysing year to year damage variability; however, since this study relies only on the mean yearly damage for estimating $T_{EOL}$, the choice of using a resampled 20-year period does not affect the results."

16) L. 354: What do you mean by "all months". I assume, all 20 SP + 10 ΔTm + all approaches (binning 1D, 2D, ... RF)? At least this is what it looks like in Fig. 6.

We thank the reviewer for this comment. Indeed, by "all months" we refer to predictions from all extrapolation approaches, across all dataset sizes, and for all randomly selected start points. To clarify, we have revised the manuscript sentence as follows:

"$T_{EOL,min}$ and $T_{EOL,max}$ represent the minimum and maximum predicted fatigue life values obtained from all extrapolation approaches, across all dataset sizes and randomly selected start points."

17) L. 374: You describe that you generate 100 random years by resampling from the 5-year period. For each year, the fatigue life is calculated. In my opinion, this leads to too high uncertainty, as the fatigue life calculation is based on a single year. In my opinion, you should use 100 random 20-year periods to be consistent with the one 20-year period used for the extrapolation methods.

We thank the reviewer for this constructive comment. We agree that using single-year resampled periods introduces noticeably higher uncertainty, as the resulting bounds represent the variability in fatigue life estimates obtained from linear extrapolation (0-d binning) across multiple individual 1 year periods rather than the variability of multiple 20-year periods. In contrast, when multiple synthetic 20-year periods are generated

through resampling from 5-year monitoring period, the resulting fatigue life estimates converge closely, producing very narrow uncertainty bounds that essentially coincide with the Mean Fatigue Life $T_{EOL}$ line in Fig. 6. In addition, while we understand the suggestion we could argue that still we are not looking at the actual uncertainty in a twenty year life as the data still only reflects 5 years of operation.

To clarify that we use the annual uncertainty as a 'reference', in the manuscript, we have added the following text:

"The uncertainty bounds shown in Fig. 6 represent the variability in fatigue life estimates obtained from linear extrapolation (0-D binning) across multiple individual 1 year periods, rather than uncertainty across multiple 20-year spans. When multiple synthetic 20-year periods are generated, the resulting fatigue life estimates show minimal spread and converge to the Mean Fatigue Life $T_{EOL}$ line in Fig. 6."

18) L. 381: You state "As data availability increases, predictions from all models begin to converge, with comparable performance across methods for data sizes beyond 18 months." In my opinion, for large data sizes and this case (3MW, single sensor), RF is outperformed by the binning approaches, as the uncertainty of RF is higher. It is totally fine that RF is outperformed by the binning approach for data sizes beyond 18 months. However, we should not present the ML approach in an overly positive light. For the 9MW turbine (l. 387), I agree with your statement that the difference diminishes.

We thank the reviewer for this comment. We agree that for the 3 MW OWT, the RF model exhibits higher uncertainty than the binning-based approaches once sufficient data are available. Our intention was not to overstate the performance of the ML model, and we have refined the text to more accurately reflect the observed behaviour. Specifically, we now highlight that although all methods trend toward convergence with increasing data availability, the higher-dimensional binning approaches show clearly lower scatter for larger data sizes, and the RF model is outperformed beyond approximately 18 months of data.

The manuscript text has been revised as follows:

"As data availability increases, predictions from all models begin to converge. Higher dimensional binning, particularly 5D binning, provides consistently lower scatter in predictions from 9 months onwards, but the mean values of predictions are shifted because of unoptimized bin sizing and bin filling strategy. The RF-5D model shows higher uncertainty and is outperformed by the binning approaches for data sizes greater than 18 months."

19) L. 393-399: In my opinion, a discussion of the higher IQR of RF compared to the binning approach for the wind speed bin 21-22 m/s is missing. This higher IQR is probably also the reason why RF is outperformed by the binning approach in Fig. 6 (see comment 18)

We thank the reviewer for this comment. We agree that the higher IQR of the RF model in the 21–22 m/s wind-speed bin requires discussion, as it directly contributes to the larger overall uncertainty of the RF predictions and explains why the RF-5D model is outperformed by the binning approaches for larger data sizes (see also Comment 18).

We have added the following explanation to the manuscript:

"As shown in Fig. 8 (right), the IQR in the wind speed bin $[21.0, 22.0)$ is consistently higher for the RF model than for the binning approach once more than 15 months of data are available. This increased scatter reflects the limited data availability in high-wind conditions. Consequently, for data sizes beyond approximately 15 months, the RF-5D model is outperformed by the binning-based approach, consistent with the trends observed in Fig. 6."

20) L. 404: You state: "However, the magnitude of these fluctuations remains relatively low." Fig. 9 has no scale on the axis. Hence, it is hard to judge how high the fluctuations are. Nonetheless, if you look at the 5-95 percentile, it looks as if RF is not well performing for this wind speed bin (Fig. 9b, 25-26m/s).

We thank the reviewer for this comment. We agree that the lack of axis scaling in Fig. 9 makes it difficult to directly assess the magnitude of the fluctuations. The trends, however, can still be compared qualitatively through the relative sizes of the IQR and 5-95 percentile bands. In the high wind speed bin [25-26 m/s], both the RF and binning methods exhibit noticeable variability, which is expected due to the very low occurrence probability and thus limited data availability in this bin. As a result, small changes in the sampled data can lead to alternating behaviour where RF shows higher IQR for some data sizes (for 6 and 12 months) and lower IQR for others (for 9 and 15 months).

We have clarified this behaviour in the manuscript with the following text:

"As shown in Fig. 9 (right), the IQR in the wind speed bin $[25.0, 26.0)$ fluctuates between the RF-5D model and the binning approach across different data sizes. For 6, 12, and 18 months of data, the binning IQR is lower, whereas for 9 and 15 months, the RF-5D model exhibits slightly lower variability. These minor shifts arise from the very limited data available in this high wind speed bin, where low occurrence probability causes this apparent variability."

21) L. 425: Again, I think that you highlight the benefits of RF too much. Not only "RF models maintain consistent and reasonable performance, particularly at smaller data sizes of 6–9 months" but 1D and 2D bins as well.

We thank the reviewer for this comment. We agree that the previous phrasing placed too much emphasis on the RF model. To better reflect the results, we have revised the statement to include the performance of the lower-dimensional binning approaches as well:

"In contrast, both lower dimensional binning (1D and 2D) and RF models show consistent and comparable performance, particularly for data sizes greater than 6 months."

22) L. 435: "For both FA and single sensor targets, the 10-90th percentile of the prediction errors stabilizes after approximately 9 months of training data. In contrast, predictions for the SS direction exhibit continued variability." Where do I see this difference in stabilisation? I assume that the low value at 12 months (preventing a good stabilisation) is just a statistical artifact/outlier

We thank the reviewer for this helpful comment. In our results, the 10-90$^{th}$ percentile range for the SS direction decreases up to 12 months and then increases again. By describing the SS direction as exhibiting "continued variability," we intended to refer to these fluctuations across different data sizes.

Considering the low value observed at 12 months as a statistical artifact, the overall trend indicates that the SS prediction errors also stabilize after approximately 9 months of data, similar to the FA and single-sensor cases.

Accordingly, we have revised the sentence in the manuscript as follows:

"For both FA and single-sensor targets, the 10-90$^{th}$ percentile of the prediction errors stabilizes after approximately 9 months of training data. For the SS direction, the percentile range shows a notably low value at 12 months. Considering the low value at 12 months a statistical outlier, the SS prediction errors also stabilize after approximately 9 months of data size."

23) L. 449: The statement "Additionally, the scatter in predictions is significantly lower for RF models than for binning-based extrapolation." is only true for short measurement periods. For longer ones, the scatter for RF models is sometimes even higher. Again, I do not say that RF has no clear advantages, but we should not present it in an overly positive light

We thank the reviewer for this helpful comment. We agree that the reduced scatter of RF predictions is primarily evident for shorter monitoring periods, and that for longer data

sizes RF can exhibit comparable or even higher scatter than binning approaches. To provide a balanced interpretation, we have revised the statement in the manuscript as follows:

"Additionally, the scatter in predictions is significantly lower for RF models than for binning-based extrapolation for smaller data sizes. For larger data sizes, for example, in the 3 MW OWT case beyond 18 months, the RF models show comparable or higher scatter and are outperformed by binning-based extrapolation models."

24) L. 454: You state that effects of the "starting point of the monitoring period" can by analysed. This is true but I would be careful with the statement, as, in my opinion, such an analysis has not be done in this work. You analysed the influence of the measurement length but did not check whether a starting point in winter/summer makes a difference.

We thank the reviewer for this comment. We agree that seasonality effects associated with choosing a starting point in specific season (e.g. winter/ summer) are not explicitly analysed in this work. However, the variability in predicted fatigue life $T_{EOL}$ observed in our results is influenced by the use of randomly varying starting points (SP) across different monitoring periods $\Delta T_m$. To avoid overstating the scope of our analysis, we have revised the sentence in the manuscript as follows

"The availability of long-term monitoring data enables the investigation of how the randomly varying starting point of the monitoring period influences the variability in fatigue life estimates."

25) L. 497: You state that "2D binning outperforms 1D binning". This has not been discussed anywhere in the paper and is not clearly visible in the data. I recommend to either discuss it somewhere else using the available figures or to remove this statement in the "conclusions"

We thank the reviewer for this comment. We agree that the statement "2D binning outperforms 1D binning" was not explicitly discussed in the manuscript. Following the reviewer's recommendation, we have removed this statement from the conclusions.

The sentence has been rephrased to:

"Higher-dimensional binning, 3D and above, suffers from issues with empty bins, especially when using smaller datasets."

Typos etc.:

26) L. 122: Remove space after "Objective section ".

27) L. 140: Remove space after "SCADA/ ".

28) Be consistent when writing "10-minute data" etc. Sometimes you write it with and sometime without hyphen.

29) L. 180: Remove space before ( Hubler and Rolfes (2022)).

30) Table 1: I would use "s" and not "sec" and "°" and not "Deg"

31) L. 342: "the model comparison" and not "the models comparison"

32) L. 344: LFFDfactor and not only LFFD

33) L. 350: Be consistent when using multiplication signs (· or preferable ×)

34) Table 3: Be consistent with the spelling of "windspeed/wind speed"

35) Table 3: "Turbine" should not be capitalized.

36) L. 431: Be consistent with the capitalisation of "Section" etc. and their abbreviations (e.g., Fig. vs Figure)

The WES submission guidelines recommend:

"The abbreviation "Fig." should be used when it appears in running text and should be followed by a number unless it comes at the beginning of a sentence, e.g.: "The results are depicted in Fig. 5. Figure 9 reveals that…"

We have revised the manuscript for consistency of capitalisation and abbreviations.

37) Table B2 and B3: Remove the * from "Operational State*" or state what it means.

We thank the reviewer for identifying all these typing and consistency issues. The manuscript has been revised accordingly.

---

## Author Comment (AC2)

Dear Reviewer,

We sincerely thank you for the positive and constructive feedback. We greatly appreciate your recognition of the relevance of our study, as well as your acknowledgement of the clarity and significance of our objectives and the contribution this work makes to the current state of the art in fatigue assessment of offshore wind turbine substructures.

Your comments were highly insightful and have helped us enhance the clarity, completeness, and scientific quality of the manuscript. We have carefully addressed each comment and have revised the manuscript accordingly, providing additional explanations, clarifications, and references where necessary.

We genuinely appreciate the time and effort you have devoted to reviewing our work.

In our response:

- The reviewer comments are in Black and
- Author replies are in Red, and
- The changes made to the manuscript are marked in Blue

The topic of the paper is very relevant, and the approach followed—anchored in two large datasets of experimental data collected from two operating wind turbines and exploring many alternative models—is quite valuable.

The objectives of the paper are clearly stated and pertinent. As the authors state at the end of the paper, some of the conclusions might depend on the site and turbine model, but still, the in-depth analysis and the results obtained for two significantly different offshore models represent an important contribution to the current state of the art.

The following points could be improved:

- **Section 3.1** – In the strain data processing, it is relevant to mention that before converting strains to stresses, it is crucial to remove the effects of temperature (this is not mentioned in the paper, but there is probably a temperature sensor next to each strain gauge) and any potential strain drift over time (these are more critical in electrical strain gauges; the paper does not specify whether the strain gauges are electrical or fiber-optic sensors).

  We thank the reviewer for this comment. The monitoring system uses electrical strain gauges, each equipped with an adjacent thermocouple for temperature compensation. Potential drift in strain measurements over time is mitigated through periodic recalibration of the strain gauges data.
  In response, we have added the following clarification to the manuscript:

*"The data from six circumferential electrical strain gauges is pre-processed using the steps shown in Fig. 3. Each strain gauge is installed together with a dedicated thermocouple to enable temperature compensation. This temperature compensation as well as any long-term measurement drift is addressed through continuous follow-up and periodic recalibration of the strain gauges data by the SHM hardware supplier."*

- **Figure 3** – In the strain pre-processing, it would be preferable to convert measured stresses to bending moments (this needs to be explained, since with the use of six measuring points, a fitting procedure should be devised). These could then be oriented in the compass direction or in the FA/SS direction, and from these, the stresses at any point of the cross-section could be obtained. The naming "stresses in FA and SS direction" is misleading—the stresses under analysis are vertical!

We thank the reviewer for this comment. We agree that the conversion from measured strains to bending moments requires explanation. We also acknowledge that the naming "stresses in FA and SS direction" may be misleading, since the stresses of interest are axial stresses resulting from FA and SS bending moments rather than stresses acting in horizontal directions.

To address this, we have added the following detailed clarification in the manuscript:

*"From the measured strains at the six circumferential sensors, the corresponding stresses and bending moments are obtained through a two-step procedure. First, the axial stress at each sensor location is computed using Hooke's law (Equation (1)):*

$$\sigma_{zz,j} = E\ \varepsilon_{zz,j}$$

*where E is Young's modulus, and $\varepsilon_{zz,j}$ and $\sigma_{zz,j}$ are the measured axial strain and resulting axial stress at the j-th sensor, respectively.*

*The general equation for normal stress $\sigma_{zz,j}$ induced by a normal force $F_N$ and the global bending moments $M_{NS}$ (North-South) and $M_{EW}$ (East-West) in cylindrical coordinates is given in Equation (2):*

$$\sigma_{zz,j} = \left(\frac{F_N}{A}\right) + R_i \cdot \left[\frac{M_{NS}}{I_C}.\sin\left(\theta_j\right) - \frac{M_{EW}}{I_C} \cdot \cos\left(\theta_j\right)\right],$$

*where A is the cross-sectional area, $R_i$ is the inner radius at the sensor location, $I_C$ is the area moment of inertia, and $\theta_j$ is the clockwise angular position of the j-th sensor from the North-South axis. Equation (2) can be written for each sensor. With six sensors, this formulation yields an overdetermined system that is solved using a least-squares fitting procedure to estimate the normal load $F_N$ and bending moments in both directions $M_{NS}$ (North-South) and $M_{EW}$ (East-West)*

*from the measurements (see Sadeghi et al. (2023a) and Link and Weiland (2014) for more details for more details).*

*The resulting bending moments are then rotated in fore-aft (FA) and side-side (SS) direction using Equation (3):*

$$\begin{Bmatrix} M_{FA} \\ M_{SS} \end{Bmatrix} = \begin{bmatrix} \cos(-\psi + \pi) & \sin(-\psi + \pi) \\ -\sin(-\psi + \pi) & \cos(-\psi + \pi) \end{bmatrix} \begin{Bmatrix} M_{NS} \\ M_{EW} \end{Bmatrix}$$

*Where $\psi$ is the mean yaw angle within each 10-minute interval. This transformation yields the fore-aft ($M_{FA}$) and side-side ($M_{SS}$) bending moments , also defined as normal bending moment ($M_{tn}$) and lateral bending moment ($M_{tl}$) respectively in IEC61400-13 (2021).*

*Stresses in fore-aft direction refer to axial stresses caused by fore-aft bending moment ($M_{FA}$) and stresses in side-side direction refer to axial stresses caused by side-side bending moment ($M_{SS}$)."*

- **Equation (1)** – In design codes, the fatigue of steel elements is calculated using a bi-linear S-N curve with m equal to 3 and 5. This should be commented on.

We thank the reviewer for this comment. We have clarified in the manuscript that design standards prescribe a bilinear S-N curve with slopes 3 and 5 for welded steel details. We have added the following description in the paper:

"The slope (m) and intercept ($\log(\bar{a})$) defining the S-N curve depend on the fatigue detail, environmental conditions, and material properties, and are provided in DNV-RP-C203 (2024) guidelines. In this study, the bilinear DNV D-A curve for a D-detail (circumferential butt weld made from both sides) is adopted, with slopes of m = [3, 5] and intercepts of $\log(\bar{a})$ = [12.164, 15.606], with a transition at $10^7$ cycles."

- **Section 3.2** – The following sentence is unclear: "Invalid operational states refer to intervals lacking valid SCADA-derived statistics, typically involving transient events such as rotor start-up or shutdown." Transient events are not considered in fatigue accumulation? Please clarify this point.

We thank the reviewer for this comment. In our workflow, each 10-minute data block is assigned to an operational state based on SCADA-derived statistics. Blocks that occur during transient events (e.g. start-up, shutdown) do not satisfy the thresholds used to classify predefined operational states and therefore receive the label *"Invalid" referring to "transient" operational state*. This label refers only to the *operational state classification* and does not imply that the data are excluded from fatigue analysis. All data intervals, including those in intervals labelled *"Invalid"*, are included in the fatigue damage accumulation.

The manuscript has been updated to replace "invalid" with "transient" to avoid confusion.

"Intervals that occur during turbine transients (e.g. start-up, shutdown) fall outside the predefined statistical thresholds on SCADA statistics and are therefore labelled *transient*."

- **Section 5, Table 3** – Some candidates for selected variables present strong correlations. Please clarify how this correlation may have influenced the selection of the features.

We thank the reviewer for this comment. While some of the selected features exhibit correlations, the Random Forest feature importance metric inherently accounts for the contribution of each variable in reducing node impurity across all trees. Therefore, even correlated features can provide incremental predictive power, and the ranking reflects their combined impact on model performance. We have added the following clarification in the revised manuscript.

"Some of the selected features exhibit correlations. Random Forest feature importance evaluates each variable's contribution to reducing node impurity across all trees, so correlated variables can still be selected if they provide incremental predictive information (Breiman (2001)). The top five features are thus those with the highest combined predictive relevance, and their selection reflects the overall importance rather than strict independence. Future work could explore dimensionality reduction methods to explicitly account for feature correlation."

- **Section 5.1** – The most standard variables used for fatigue estimation with SCADA data are wind velocity and turbulence. It would be useful to compare a model just based on these variables with all the others that have been tested.

We thank the reviewer for this comment. While wind speed and turbulence intensity are indeed standard variables for fatigue estimation, our study focuses on identifying the most relevant features based on the Random Forest feature importance metric. Turbulence intensity is selected for nearly all target variables (except the side-side direction of the 9 MW OWT), and the binning approaches includes wind speed and turbulence intensity alongside other top-ranked features. We acknowledge the value of a comparison with a model using only standard variables; however, this is beyond the scope of the current study, which emphasizes incremental feature selection and its impact on predictive performance. We have added reference to earlier relevant work along with the following lines in the paper to address the reviewers comment:

"While wind speed and turbulence intensity are commonly used for fatigue estimation (as studied by Noppe et al. (2020) on 2MW OWT), our feature selection approach identifies variables that provide the greatest incremental predictive value for each target. Turbulence intensity is selected for most targets, and wind speed is included in all binning strategies (except for side-side direction of 9MW OWT), ensuring that standard fatigue predictors are considered alongside additional influential features."